# Investigating multiscale meteorological controls and impact of soil moisture heterogeneity on radiation fog in complex terrain using semi-idealised simulations

Dongqi Lin[1,2], Marwan Katurji[2], Laura E. Revell[1], Basit Khan[3], and Andrew Sturman[2]

[1]School of Physical and Chemical Sciences, University of Canterbury, Christchurch, New Zealand
[2]School of Earth and Environment, University of Canterbury, Christchurch, New Zealand
[3]Institute of Meteorology and Climate Research, Atmospheric Environmental Research (IMK-IFU), Karlsruhe Institute of Technology (KIT), 82467 Garmisch-Partenkirchen, Germany

**Correspondence:** Dongqi Lin (dongqi.lin@canterbury.ac.nz)

**Abstract.** Coupled surface-atmosphere high-resolution mesoscale simulations were carried out to understand meteorological processes involved in the radiation fog life cycle in a city surrounded by complex terrain. The controls of mesoscale meteorology and microscale soil moisture heterogeneity on fog were investigated using case studies for the city of Christchurch, New Zealand. Numerical model simulations from the synoptic to micro- scale were carried out using the Weather Research and Forecasting (WRF) model and the Parallelised Large-Eddy Simulation Model (PALM). Heterogeneous soil moisture, land use, and topography were included. The spatial heterogeneity of soil moisture was derived using Landsat 8 satellite imagery and ground-based meteorological observations. Nine semi-idealised simulations were carried out under identical meteorological conditions. One contained homogeneous soil moisture of about 0.31 m$^3$ m$^{-3}$, with two other simulations of halved and doubled soil moisture to demonstrate the range of soil moisture impact. Another contained heterogeneous soil moisture derived from Landsat 8 imagery. For the other five simulations, the soil moisture heterogeneity magnitudes were amplified following the observed spatial distribution to aid our understanding of the impact of soil moisture heterogeneity. Analysis using pseudo-process diagrams and accumulated latent heat flux shows significant spatial heterogeneity of processes involved in the simulated fog. Our results showed that soil moisture heterogeneity did not significantly change the general spatial structure of near-surface fog occurrence, even when the heterogeneity signal was amplified and/or the soil moisture was halved and doubled. However, compared to homogeneous soil moisture, spatial heterogeneity in soil moisture can lead to changes in fog duration. These changes can be more than 50 minutes, although they are not directly correlated with spatial variations in soil moisture. The simulations showed that the mesoscale (10 to 200 km) meteorology controls the location of fog occurrence, while soil moisture heterogeneity alters fog duration at the microscale on the order of 100 m to 1 km. Our results highlight the importance of including soil moisture heterogeneity for accurate spatiotemporal fog forecasting.

## 1 Introduction

Dense fog has been identified as one of the greatest hazardous meteorological phenomena in the atmospheric boundary layer (ABL) (Duynkerke, 1991; Mason, 1982). Fog is a result of water droplets and/or ice crystals occurring in suspension near the

Earth's surface, which limits visibility to less than 1 km (Brown and Roach, 1976; Gultepe et al., 2007; Steeneveld et al., 2015; WMO, 1992). Fog can be classified into several types based on the main driving meteorological processes (e.g., Gultepe et al., 2006; Tardif and Rasmussen, 2007; Van Schalkwyk and Dyson, 2013; Belorid et al., 2015; Bari et al., 2016; Roux et al., 2021; Lin et al., 2023a). For instance, advection fog is usually related to a moist warm air mass moving over a colder surface, while radiation fog is driven by more localised processes, particularly radiative cooling of the ground surface. The main focus of this paper is radiation fog. Over 30 years ago, Duynkerke (1991) identified land surface physical characteristics as the most important factor for fog occurrence, among several thermal and dynamical processes in the ABL. The physical characteristics of the land surface, including land use, soil moisture, and soil temperature, have a significant impact on the coupled energy and water exchanges between the land surface and ABL (e.g., Bou-Zeid et al., 2004; Maronga et al., 2014; Rihani et al., 2015; Shao et al., 2013; Srivastava et al., 2020). Gultepe et al. (2007) pointed out that radiation fog over a heterogeneous land surface is difficult to forecast. The spatial heterogeneity in land-atmosphere interactions and surface energy balance can impact energy transfer between the atmosphere, ground surface, and soil (e.g., Courault et al., 2007; Huang and Margulis, 2013; Maronga et al., 2014). Here, we recognise heterogeneity as spatial variability in land surface characteristics, while homogeneity means that such characteristics are spatially uniform.

Over the past decade, many studies have included heterogeneous land surface characteristics in radiation fog simulations in order to understand the microscale processes (occurring from 1 cm to 1 km, and from seconds to hours) and associated feedback during the fog life cycle. High resolution numerical weather models, including Large Eddy Simulation (LES) models, are useful tools for improving the understanding of dynamical processes involved in the radiation fog life cycle within a complex environment. With higher resolution, the microscale processes related to surface heterogeneities can be better resolved and represented. Bergot et al. (2015) investigated the impact of the heterogeneously built environment over flat terrain at Paris–Charles de Gaulle airport, while Mazoyer et al. (2017) demonstrated the impact of a plant canopy on a radiation fog event observed during the ParisFog field campaign (Haeffelin et al., 2010). Both studies applied simulations at grid spacings $\leq 5$ m using an LES model and found that variations in land use lead to spatial heterogeneities of fog.

While numerical simulations at metre-scale are highly valuable, they are also highly computationally expensive. In the last few years, to compromise between computational cost and simulation scale, several studies have carried out simulations over complex heterogeneous topography at sub-km scale (with grid spacing of a few hundred metres). For example, Bergot and Lestringant (2019) carried out a case study with two horizontal numerical grid spacings (500 m and 50 m) in a region of north-eastern France. Ducongé et al. (2020) conducted simulations with 100 m horizontal grid spacing in the Shropshire hills (UK), and Smith et al. (2020) applied sub-km simulations (1.5 km, 333 m, and 100 m) over the Cardington and Shropshire regions. All these studies highlighted that even over a relatively small area (less than 4 $\text{km}^2$), when a complex and heterogeneous environment is present, fog occurrence and types can vary significantly.

In addition to topography, soil moisture is identified as an important attribute of land surface characteristics. Changes in soil moisture lead to variability in the surface energy balance, and consequently changes in fog formation and dissipation times (Bergot and Guedalia, 1994; Maronga and Bosveld, 2017; Rémy and Bergot, 2009). Maronga and Bosveld (2017) investigated the impact of perturbations in soil moisture at model initialisation on radiation fog, and found that such perturbations only

affect the lifting and dissipation of the fog layer, while fog formation time was not impacted. They highlighted that drier soil was associated with more energy at the surface contributing to greater surface sensible heat flux. As a consequence of increased surface sensible heat flux, the rate of surface heating increased leading to earlier dissipation of the fog layer. However, Maronga and Bosveld (2017) only conducted their simulations over grassland with flat terrain and homogeneous soil moisture. Smith et al. (2020) included complex topography and demonstrated the impact of soil thermal conductivity and different choices of land surface parameterization on fog formation, dissipation, and duration. Temperature biases were found in the heterogeneous hill and valley structures, with simulated temperatures over hills being too cold leading to too much fog, while the temperatures over the valleys were too warm leading to too little fog. In addition, they showed that fog formation time is highly sensitive (with up to a 4 hour difference) to soil and land surface parameterizations, while dissipation time is relatively insensitive (with a 1.5 hour difference at most). However, to the best of our knowledge, no study has investigated the impact of soil moisture heterogeneity on radiation fog in combination with heterogeneous land use and topography.

Despite numerous fog studies worldwide, only a small number has been carried out for Christchurch (or, for that matter, New Zealand) (e.g., Lin et al., 2023a; Osborne, 2002; Hume, 1999). Christchurch (43.5321° S, 172.6362° E) is located on the Canterbury Plains, on the east coast of the South Island, New Zealand (Figure 1). The city experiences fog events on approximately 44-49 days per year (Macara, 2016; New Zealand Meteorological Service, 1982). As the largest city of New Zealand's South Island, it is a vital port for aviation, shipping and surface transport. Hence, an accurate fog forecast is important for Christchurch and its international airport. Although Christchurch is a moderate-sized city (area of 1,426 km$^2$), great spatial heterogeneity exists in its land surface characteristics, which makes accurate fog forecasting difficult. Mesoscale flow systems are modified by the Southern Alps (highest peak elevation of 3754 m) to the west and Banks Peninsula (highest peak elevation of 920 m) to the south. An example of the impact of local flow phenomena (in this case mesoscale in nature) is when westerly flows passing over the Southern Alps interact with the north-easterly airflow along the coast, creating a wind convergence zone over Christchurch (e.g., Corsmeier et al., 2006; Sturman and Tapper, 2006). At nighttime, local airflow is also affected by cold air drainage from both the foothills of the Southern Alps and the Canterbury Plains to the west and the Port Hills on the southern boundary of the city. In the Christchurch area, radiation fog is the predominant type of fog, and such fog events usually coincide with stagnant air zones associated with local drainage flows (Lin et al., 2023a).

In order to include forcing from local circulation and/or the terrain-induced drainage flows, the domain size of the simulations should be of the order of a few tens of kilometres (Ducongé et al., 2020), while several LES fog simulations at metre-scale have usually had a domain size less than 1 km (e.g., Bergot et al., 2015; Maronga and Bosveld, 2017; Schwenkel and Maronga, 2019). Considering the high computational cost, the optimal approach is to carry out high-resolution mesoscale simulations (Cuxart, 2015) at sub-km grid spacing. The surface and topographic heterogeneities can be partially resolved in such high-resolution mesoscale simulations, and consequently the dynamical processes and spatial variability of fog can be captured (Vosper et al., 2013, 2014). This study therefore aims to investigate the impact of soil moisture on radiation fog duration using high-resolution mesoscale simulations for Christchurch. In addition, we want to identify the meteorological controls that may be relevant for fog forecasting. The main objectives of this study are therefore to:

1. investigate the major meteorological controls on radiation fog in the complex environment of Christchurch

2. investigate the impact of soil moisture heterogeneity on radiation fog

A radiation fog scenario was created in semi-idealised numerical simulations containing nested domains with the finest grid spacing of 81 m. Data were obtained from numerous sources including meteorological observations and the Weather Research and Forecasting modelling system (WRF; http://www.wrf-model.org, last access: 17 May 2022). The fog simulations themselves were conducted using the Parallelised Large-Eddy Simulation Model (PALM) (Maronga et al., 2015, 2020). The WRF model is a state-of-the-art numerical atmospheric model that has high applicability in multiple world regions, and has been extensively used in numerous atmospheric and climate studies (Skamarock et al., 2021). PALM has been used for boundary layer studies for more than 20 years, including radiation fog studies in a stable boundary layer (Maronga and Bosveld, 2017; Schwenkel and Maronga, 2019). High resolution soil moisture data were derived from Landsat 8 imagery (at a resolution of 30 m) and used to construct experimental initial conditions of soil moisture in PALM. The experiments included a range of spatially heterogeneous soil moisture conditions in order to assess the impact of soil moisture on radiation fog. Pseudo diagrams described in Steeneveld and de Bode (2018) and Bergot and Lestringant (2019) were adopted to investigate the driving processes of the simulated fog.

## 2 Data description

Data sources used in this study are shown in Tables 1 and 2. To conduct PALM simulations, two types of input are important. One is the static driver, which includes geospatial data of land surface information and characteristics, such as topography, buildings, streets, vegetation, soil types, and water bodies. The other is the dynamic driver which contains the vertical profiles of the atmosphere and soil. The WRF simulations were initialised using data from the fifth generation of the European Centre for Medium-Range Weather Forecasts (ECMWF) atmospheric reanalysis of the global climate (ERA5) (Hersbach et al., 2019), and the WRF model configuration is identical to that described in Lin et al. (2021).

### 2.1 Geospatial data sets

As shown in Table 1, digital elevation model (DEM) and digital surface model (DSM) data were obtained from Environment Canterbury Regional Council (2020) for the major urban area of Christchurch. Both the DEM and the DSM have a spatial resolution of 1 m. The DEM provides information of ground surface altitude only, while the DSM includes the heights of all surface objects including buildings and trees. For areas outside of the major urban area, 1 m resolution DEM data are not available so we used a DEM with a spatial resolution of 25 m provided by Landcare Research (2018). No DSM is available outside of the main Christchurch metropolitan area. As we mainly focus on the area inside Christchurch, the 25 m DEM is sufficient to resolve the features and impacts of the surrounding areas. Land use information was obtained from Land Cover Database (LCDB) New Zealand (Landcare Research, 2020), while urban surface data, such as building outlines, streets and pavements were obtained from OpenStreetMap (https://planet.openstreetmap.org/; last access 14 April 2022). Land surface information and all geospatial data were included in the static driver input of PALM. In the static driver input files, the water temperature was set to 282.45 K based on the August mean of 2001 obtained from ERA5 reanalysis. The projection used for

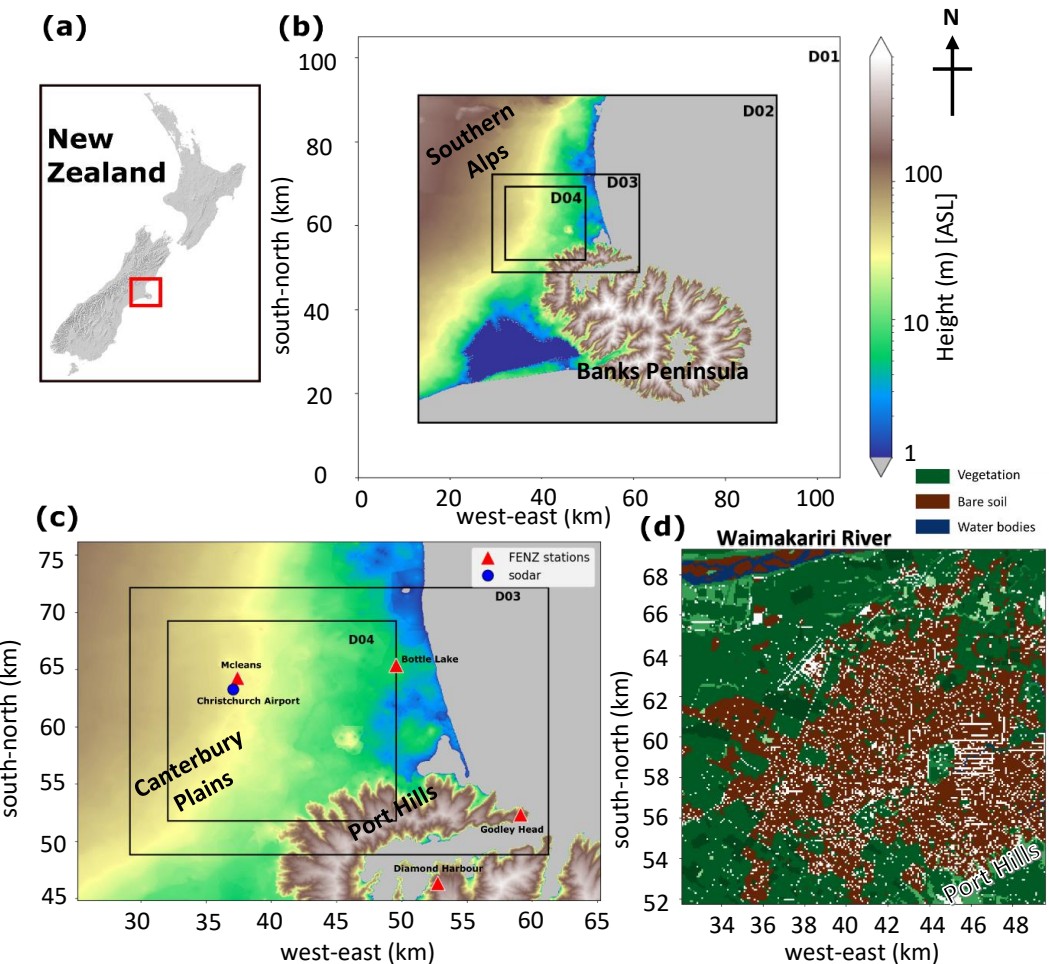

**Figure 1.** Maps of the case study simulation domains: (a) a New Zealand topographic map with a red square indicating the location of the simulation domains, (b) a topographic map (height above sea level) showing the simulation domain configuration for the case study, (c) a topographic map of simulation domains 3 and 4 (D03 and D04), the locations of the FENZ (Fire and Emergency New Zealand) weather stations, and the location of the sodar operated at Christchurch airport, and (d) a land use map of D04 with buildings and streets in white. The simulation domain 1 (D01) configured as flat terrain with short grass, and hence no topography, is shown in panel (b). The logarithmic topographic height colormap in panel (b) applies to panels (b) and (c), while grey indicates the ocean.

the static data is New Zealand Transverse Mercator (NZTM) - EPSG:2193. The static driver input files were created using a similar procedure to that described in Heldens et al. (2020), Lin et al. (2021), and Lin et al. (2023b).

**Table 1.** Sources and description of geospatial data sets used in this study.

| Geospatial data | | |
|---|---|---|
| **Data set** | **Source** | **Description** |
| Christchurch Digital Elevation Model (DEM) with spatial resolution of 1 m | Environment Canterbury Regional Council (2020) | Topographic height data for Christchurch city |
| Christchurch Digital Surface Model (DSM) with spatial resolution of 1 m | Environment Canterbury Regional Council (2020) | Data for Christchurch city, including altitude and height of surface geometries, such as buildings and trees |
| New Zealand South Island DEM with spatial resolution of 25 m | Landcare Research (2018) | Topographic height data for South Island of New Zealand |
| New Zealand land cover database (LCDB) V5.0 | Landcare Research (2020) | Land use categories for New Zealand |
| OpenStreetMap | https://planet.openstreetmap.org/ (last access: 14 April 2022) | Locations and outlines of buildings, pavements, and streets |

## 2.2 Meteorological data sets

One particular challenge of conducting fog research in Christchurch is its poor observational network. Despite being the second largest city in New Zealand, Christchurch has limited availability of meteorological observational data. Only the automatic weather station (AWS) operated at Christchurch International Airport (CHA) provides visibility measurements over the last 20 years. There are also no regular radiosonde launches in Christchurch, meaning no long-term vertical profiling data are available for investigation of the vertical structure of fog. Therefore, here we only conducted semi-idealised simulations by taking vertical profiles from both observations and WRF simulation. The scope of this study is not to replicate a real fog event. Rather, we only aim to create a radiation fog scenario to investigate the impact of soil moisture heterogeneity on radiation fog.

The vertical profiles of temperature, wind speed and direction, pressure, and measurement height were obtained from the national climate database (CliFlo; https://cliflo.niwa.co.nz/) operated by the National Institute of Water and Atmospheric Research (NIWA). These upper air measurements were recorded by sensors on aircraft arriving at and departing from Christchurch airport. The measurement frequencies for temperatures and winds are every 12 hours and 6 hours, respectively, when aviation is not impacted by weather conditions. Water vapour mixing ratio data were obtained from WRF simulations as these are not available from the observations. For more details of the observed fog event mentioned in this study, readers are referred to Appendix A.

**Table 2.** Sources and description of meteorological data sets used in this study.

| Meteorological data | | |
|---|---|---|
| **Data set** | **Source** | **Description** |
| ERA5 | ECMWF | Meteorological input data for WRF simulation and water temperature in PALM simulations |
| Upper air observations | CliFlo (https://cliflo.niwa.co.nz/; last access: 21 October 2022) | Vertical profiles of wind, temperature, pressure, and measurement heights for PALM initialisation |
| Sodar observations | University of Canterbury, New Zealand | Radiation fog event identification and verification |
| Landsat 8 satellite soil observations | Sentinel Hub EO Browser (Sinergise Ltd, 2022a, b, last access: 14 April 2022) | Landsat 8 satellite imagery providing the pattern of soil moisture heterogeneity |
| Ground-based soil moisture content observations | New Zealand Modelling Consortium, Open Environmental Digital Library (https://envlib.org; last access: 21 October 2022) | Soil moisture content observational data obtained from Automatic Weather Stations (AWSs) operated by Fire and Emergency New Zealand (FENZ) |

In terms of the soil moisture data set, we do not aim to reproduce the soil moisture profiles during the selected radiation fog event. Rather, satellite observations are used as a tool to derive spatial patterns of soil moisture heterogeneity that can be included in the simulations to investigate its impact on the radiation fog life cycle. Based on data availability, Landsat 8 data were used in this study to provide high-resolution soil moisture observations. The moderate resolution imaging spectroradiometer (MODIS) is another potential data source, but MODIS only provides data at resolutions between 250 m and 1 km (Justice et al., 2002). In contrast, Landsat 8 has a spatial resolution of 15 m to 100 m (Roy et al., 2014), and the surface soil moisture heterogeneity can be well-captured with such a high spatial resolution. Estimation of soil moisture from satellite observations requires ground-based measurements of surface soil moisture content. In New Zealand, such observations are only available from the AWSs operated by Fire and Emergency New Zealand (FENZ; locations see Figure 1), and most soil sensors were only deployed at the AWSs in Christchurch after 2018. Consequently, Landsat 8 observations from 2nd August 2019 were used to derive the heterogeneously distributed soil moisture profile for this case study, and were obtained from Sentinel Hub EO Browser (Sinergise Ltd, 2022a, b, last access: 14 April 2022).

There are several reasons why this day was chosen. First, a winter case was selected so that the soil moisture data represent winter soil conditions over Christchurch. Second, cloud coverage varies with every satellite pass, and for 2nd August 2019 only a few clouds were present when the measurements were made. Therefore, surface soil moisture can be obtained for the entire simulation domain for this day. In addition, no significant precipitation or drought event occurred prior to 2nd August 2019. The derived values of soil moisture fall within the range of long-term measured minimum and maximum values for the Canterbury region, as described in Sohrabinia et al. (2014). This means that the derived soil moisture is suitable for conducting simulations representing winter conditions. Furthermore, the surface soil moisture values observed by the four FENZ stations (locations shown in Figure 1c) are similar to those obtained from the WRF simulations (not shown), and only minor adjustment is needed to include the derived soil moisture heterogeneity in the PALM simulations.

## 3 Model and simulation configuration

The non-hydrostatic PALM model system 6.0 (r4829) was used to conduct the fog simulations. The PALM simulation configuration is shown in Table 3. All simulations used a modified three-dimensional Deardorff 1.5-order turbulence closure scheme, in which the energy transport by sub-grid scale eddies is assumed to be proportional to the local gradient of the average quantities (Deardorff, 1980; Moeng and Wyngaard, 1988; Saiki et al., 2000; Maronga et al., 2015). Considering the order of magnitude of their sizes, most of the eddies (over 60%) are parameterized in the simulations presented in this study (which used 81 m horizontal grid spacing and 18 m vertical grid spacing). Therefore, following the terminology discussed by Cuxart (2015), our simulations are high-resolution mesoscale simulations rather than LES, despite using the LES model PALM. The simulation includes four model domains (Figure 1) using a one-way nesting mode. Figure 2 shows the vertical profiles of the atmosphere and soil used for model initialisation. The vertical profiles of potential temperature ($pt$), west-east component of wind ($u$), and south-north component of wind ($v$) were obtained from the upper air measurements. In addition, data from WRF simulations were used to derive the vertical profiles of water vapour mixing ratio ($qv$), soil moisture, and soil temperature. These data obtained from WRF are not available from the observational network of Christchurch. The initialisation time was 0000 local standard time (LST) 5th August. The soil moisture and soil temperature profiles were derived from the mean values of the WRF grid points within the PALM simulation domain 4 (D04). The vertical profiles of soil moisture and soil temperature shown in Figures 2e and 2f are used for the homogeneous setup (hereafter HOM), with the values of soil moisture and temperature identical horizontally for each soil layer. To include the spatial heterogeneity of soil moisture, a 3D field of soil moisture (west-east, south-north, and eight vertical soil layers) was used at initialisation. For the heterogeneous setup (hereafter HET), the vertical profile of soil moisture was adjusted corresponding to the surface soil moisture heterogeneity derived from Landsat 8. The vertical profiles obtained from the upper air measurements and WRF were processed to create the dynamic driver input files for PALM by using the initialisation profile interpolation functions applied in WRF4PALM (Lin et al., 2021).

This study does not aim to accurately reproduce the entire radiation fog event. Rather, the aim is to create a radiation fog scenario for a stable boundary layer and use the simulations to provide guidance on the processes involved in the radiation fog life cycle. Therefore, the dynamic driver is only used to initialise the simulations and cyclic lateral boundary conditions are

**Table 3.** Configuration of PALM simulation.

| Domains | D01 | D02 | D03 | D04 |
|---|---|---|---|---|
| **Grid spacing west-east (dx)** | 729 m | 729 m | 243 m | 81 m |
| **Grid points west-east (nx)** | 144 | 108 | 132 | 216 |
| **Grid spacing south-north (dy)** | 729 m | 729 m | 243 m | 81 m |
| **Grid points south-north (ny)** | 144 | 108 | 96 | 216 |
| **Vertical grid spacing (dz)** | 162 m | 162 m | 54 m | 18 m |
| **Vertical grid points (nz)** | 32 | 18 | 24 | 36 |
| **Topography** | Flat | Derived from DEM | Derived from DEM | Derived from DEM |
| **Radiation model** | RRTMG | RRTMG | RRTMG | RRTMG |
| **Land use** | short grass | Different types of vegetation, soil, and water bodies, derived from New Zealand LCDB v5.0 | Different types of vegetation, soil, and water bodies, derived from New Zealand LCDB v5.0 | Different types of vegetation, soil, and water bodies, leaf area density, plant canopy, buildings, streets, and pavements, derived from New Zealand LCDB v5.0 and OpenStreetMap |
| **Bulk cloud model** | Off | Off | Off | On (Kessler) |
| **Soil moisture** | Homogeneous | Homogeneous | Homogeneous | Heterogeneous |
| **Boundary conditions** | Cyclic | Nested | Nested | Nested |

used to drive the model. All fog simulations in this study have a 48 hour run time and the first 24 hours of the simulations are considered as model spin-up time. Only the second 24 hours of the simulations are therefore included in the analysis. As previously mentioned, the first domain (D01) is configured as flat terrain with the land use type configured as grassland only. This is because grassland is the dominant land use type in the rural area surrounding Christchurch. The purpose of this domain is to pass down the synoptic forcing to the finer domains and and to avoid numerical instability caused by steep terrain near the periodic lateral boundaries. The heterogeneous terrain and land use are enabled for the nested domains to represent the

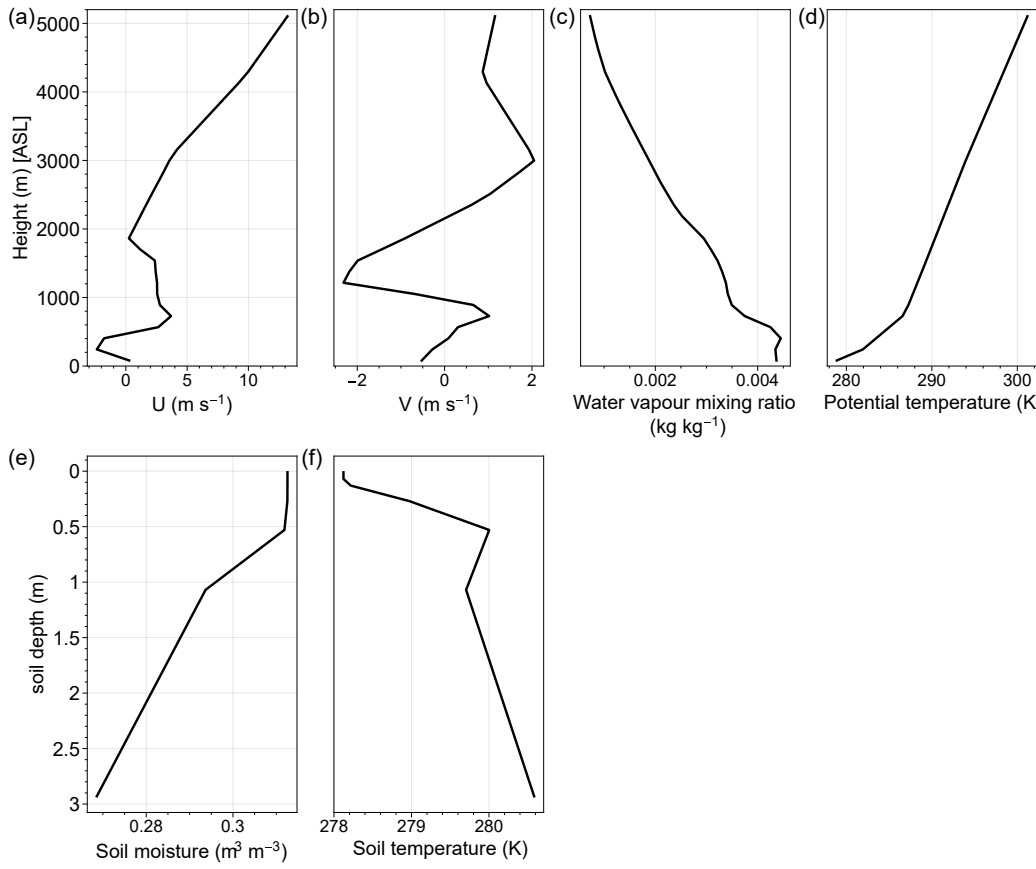

**Figure 2.** Vertical profiles at simulation initialisation: (a) $u$ wind component, (b) $v$ wind component, (c) water vapour mixing ratio ($qv$), (d) potential temperature ($pt$), (e) soil moisture, and (f) soil temperature. Note that the profile of soil moisture shown here was only applied in the simulation with homogeneous soil moisture.

complex rural-urban environment. We focus on D04 to investigate the impact of soil moisture heterogeneity on radiation fog. Therefore, the urban surface model, plant canopy model, and the bulk cloud model are only enabled for D04. In domains 2 and 3 (D02 and D03, respectively), different types of vegetation, soil, and water bodies were included (as described in Table 3). Only D04 includes leaf area density, plant canopy, and urban canopy (buildings, pavements, and street). The bulk cloud model was switched off for domains D01, D02, and D03, to simplify the processes involved in the simulated fog.

In the fog simulations, the land surface model (Gehrke et al., 2021), urban surface model (Resler et al., 2017), radiation model (RRTMG) (e.g., Clough et al., 2005), plant canopy model (Maronga et al., 2020), LES-LES nesting, and one-moment bulk cloud model (Kessler, 1969) are used. Two-moment schemes are not compatible with the plant canopy model of PALM. Cloud water sedimentation based on Ackerman et al. (2009) is enabled. Fog is identified when liquid water is present at the first model level, i.e. when liquid water mixing ratio ($ql$) is greater than zero. Due to the inclusion of the plant canopy model

and bulk cloud model in PALM, the three-dimensional Radiative Transfer Model (RTM; Krč et al., 2021) was switched off. The surface radiation transfer is then directly computed by the RRTMG model embedded in PALM. This configuration of the radiation model in PALM fog simulations is similar to those described in Maronga and Bosveld (2017) and Schwenkel and Maronga (2019). For more detailed technical descriptions of PALM, the reader is referred to Maronga et al. (2015), Maronga and Bosveld (2017), Maronga et al. (2020), and Krč et al. (2021).

## 4  Meteorological controls in the baseline scenario

In this section, the meteorological controls involved in the simulated fog event are presented and discussed. We only present the results for D04 obtained from HOM as a reference, as the initialisation profiles of the atmosphere used for all the simulations are identical. Figure 3 shows the fog duration, $qv$, and the modified Richardson number (MRi) at the lowest model level. In Figure 3, $qv$ and MRi were obtained at the time when fog was first detected at each grid point. The MRi is calculated as:

$$MRi = \frac{T_{air} - T_{sfc}}{u^2} \qquad [\text{K m}^{-2} \text{ s}^2] \tag{1}$$

where $T_{sfc}$ is surface temperature, and $T_{air}$ and $u$ are temperature and wind speed at the first model level (9 m above the ground surface), respectively. Here the MRi was used to assess the dynamic stability of the near-surface layer (Baker et al., 2002; Lin et al., 2023a). MRi > 0 indicates a stable surface layer with a surface temperature inversion. The more positive the MRi, the stronger the near-surface temperature inversion and the smaller the wind speed. MRi < 0 indicates that the surface layer is unstable, with the surface warmer than the overlying air so that turbulence is not suppressed.

We focus on four sites in D04 where fog events are the most recognisable and have a relatively long duration. The four sites are marked in Figure 3 and are noted as Hagley Park (HAP), Port Hills (PTH), Southwest of Christchurch (SWC), and Waimakariri River (WMR). We recognise fog occurred when $ql$ is detected at a grid point at the lowest model level. The study period is between 2000 LST 5th August 2001 and 1100 LST 6th August 2001. As the presented simulations here are semi-idealised, we hereafter refer to 5th August as Day 1 and 6th August as Day 2. In general, fog forms at locations with either high positive MRi (strongly stable) or relatively high $qv$ (> 5.0 g kg$^{-1}$). This agrees with known characteristics of radiation fog that sufficient radiative cooling and moisture availability play major roles in its formation and development (e.g., Duynkerke, 1991; Gultepe et al., 2007).

Significant spatial variation of fog occurred in the simulation and several meteorological controls appeared to impact on fog formation, development and dissipation. To provide more insight into the fog simulation, the spatial distribution of $ql$ and MRi on the first model level for D04 between 0030 LST and 0830 LST are shown in Figures 4 and 5. One-hour averages were applied to the horizontal cross sections. The sunset time for this case study was around 1725 LST on Day 1. Sunrise is around 0734 LST on Day 2. In addition, the one-hour averaged vertical cross sections over central Hagley Park of $pt$, $qv$, and $ql$ between 0100 LST and 0700 LST are shown in Figure 6. The south-north vertical cross sections were obtained at the 150th grid point along the west-east axis (approximately 12.2 km, location shown in Figure 3a).

Fog formation started around HAP where the temperature decreased faster than its surroundings (Figures 4a and 6a). The area within 1 km centred at HAP is located in the low-lying centre of the city (at about 9 m above mean sea level) and is

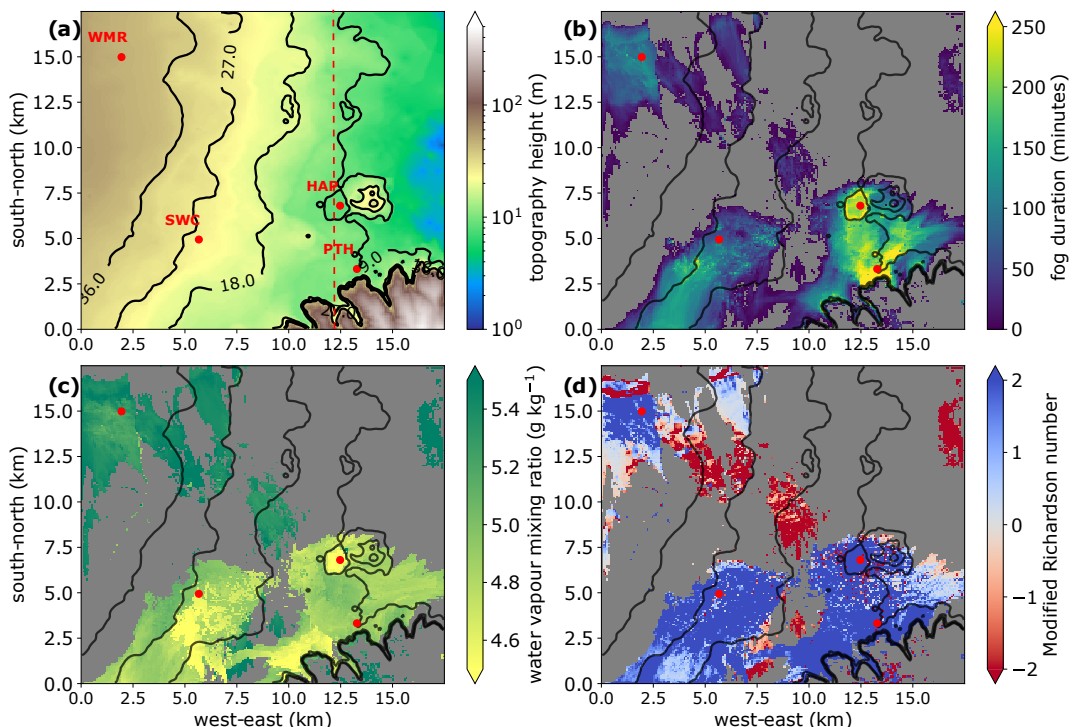

**Figure 3.** (a) Topographic map of D04 with black contour lines indicating terrain height between 9 m and 36 m. (b) Fog duration in minutes in the HOM simulation. Panels (c) and (d) show water vapour mixing ratio and the modified Richardson number (MRi) when fog was first detected at the grid cell at the first model level for D04 in HOM, respectively. Contour lines in panels (b), (c), and (d) indicate terrain height labelled in panel (a). Grey areas in (b), (c), and (d) are where no fog occurred. Hagley park (HAP), Port Hills (PTH), Southwest of Christchurch (SWC), and Waimakariri River (WMR) are marked by red dots in each panel. The red dashed line in panel (a) indicates the location of the vertical cross sections shown in Figure 6.

sheltered relative to the surrounding area (Figure 3a). During the entire simulation period, the surface wind speed over this area did not exceed 1.5 m s$^{-1}$, remaining well below 0.5 m s$^{-1}$ during night time. The low wind speed enhances the sheltering effect, and the flow within the low-altitude area of HAP became decoupled from the flow outside. A stable layer therefore started to form with reduced downward turbulent heat flux. The stable near-surface layer over HAP is highlighted by the high positive MRi shown in Figures 5a-c. While the stable layer and fog were developing, the northwesterly flow coming across the Canterbury Plains from the foothills of the Southern Alps and the southeasterly flow coming from the Port Hills were converging across Christchurch. The convergence zone is visible in the wind vector fields shown in Figures 4a-c, which agrees with the wind field described in Corsmeier et al. (2006).

Between 0100 LST and 0300 LST, as the stable layer developed further around HAP, fog started to form and develop in the surrounding areas (Figures 4a-d). During this period, the northern part of D04 remained less stable coinciding with a relatively strong (1.5 m s$^{-1}$) northwesterly flow. This flow lifted above the cold air drainage from the Port Hills, the stable layer, and

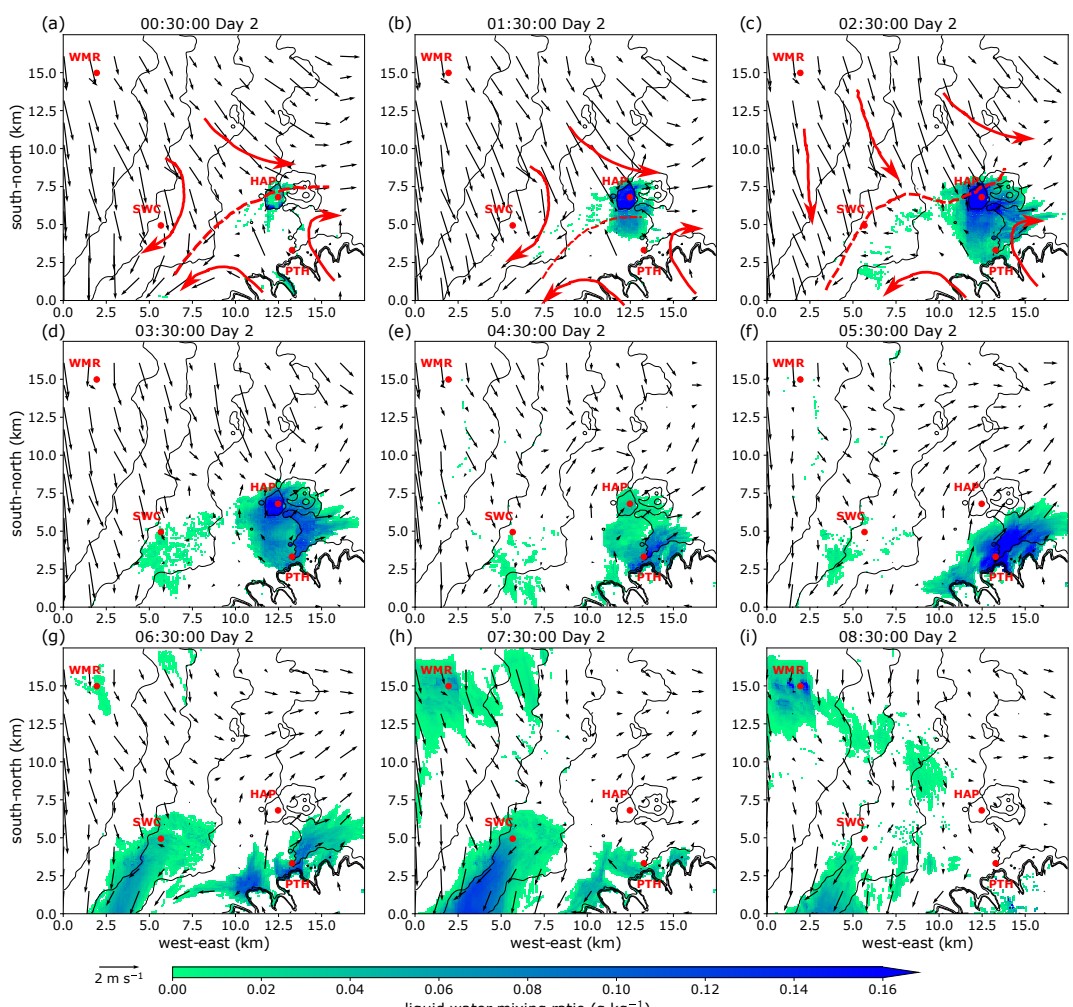

**Figure 4.** Lquid water mixing ratio ($ql$) and wind vectors at the first model level between 0030 LST and 0800 LST of Day 2. An interpretation of the drainage flow (solid red arrows) and the convergence zone (dashed red lines) is shown in panels (a)-(c). Grey contour lines in each panel indicate terrain height labelled in Figure 3a. The four sites of interest are marked by red dots.

the fog top over HAP (Figure 6j), thereby controlling the fog thickness. Consequently, the fog layer over HAP was unable to develop above approximately 100 m. The interaction between the relatively strong northerly flow and the fog top is considered to be an important factor leading to fog dissipation over HAP. Vertical entrainment appears to have resulted in turbulent mixing at the fog top over this area, while the northerly flow pushed the convergence zone to the south. As a result, fog dissipated over HAP and its surrounding areas and started to retreat to the south along the base of Port Hills (Figure 4d-g). The fog kept developing in pockets along the edge of the Port Hills where the surface layer was highly stable in association with high water vapour mixing ratio near the surface (Figures 3e-g, and 4e-g). The fog was not sustained over the valleys where the flows were channelled down from the Port Hills (Figure 4e-h).

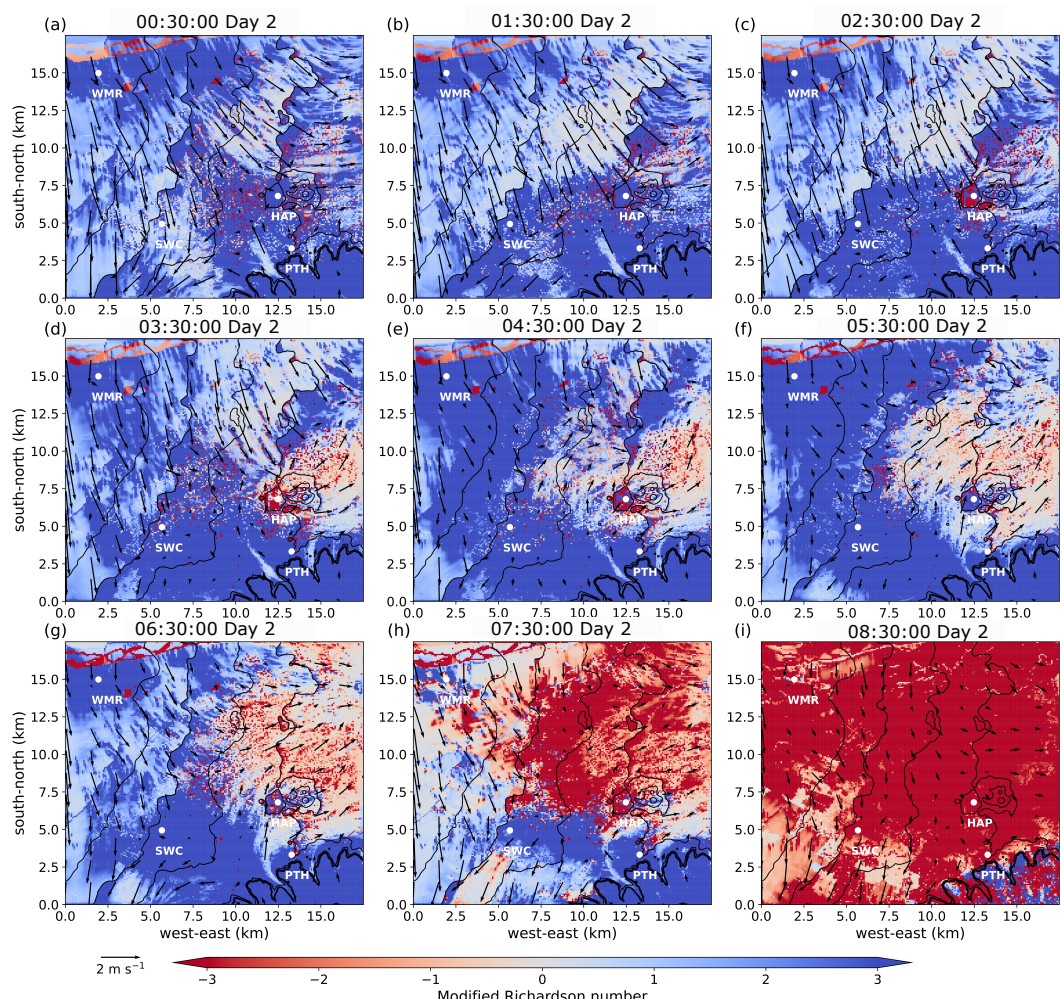

**Figure 5.** Similar to Figure 4, but for the modified Richardson number (MRi) and wind vectors. The four sites of interest are marked by white dots.

At around 0630 LST, the southeasterly flow from the Port Hills converged with the northerly flow near SWC. Another cold air drainage zone developed due to the flow convergence, where the turbulence was suppressed and a stable layer started to grow. Figures 4e-g shows patchy fog appeared around SWC. The fog was later advected further to the southwest of D04 by the converged flow turning to become northeasterly. In addition, one can notice that another fog event occurred near WMR (Figures 4h-i), forming at around 0730 LST. This was considered as a result of water vapour evaporating from the Waimakariri River located to the northwest of Christchurch and condensing into liquid water. High $qv$ was present over the Waimakariri River at fog formation (Figure 3c). The duration of these two fog events later in the morning was relatively short as they both occurred near sunrise and dissipated soon after due to an increase in incoming shortwave radiation and turbulent mixing. After sunrise, the stable near-surface layer over the entire domain decayed and became unstable (Figures 6g-i).

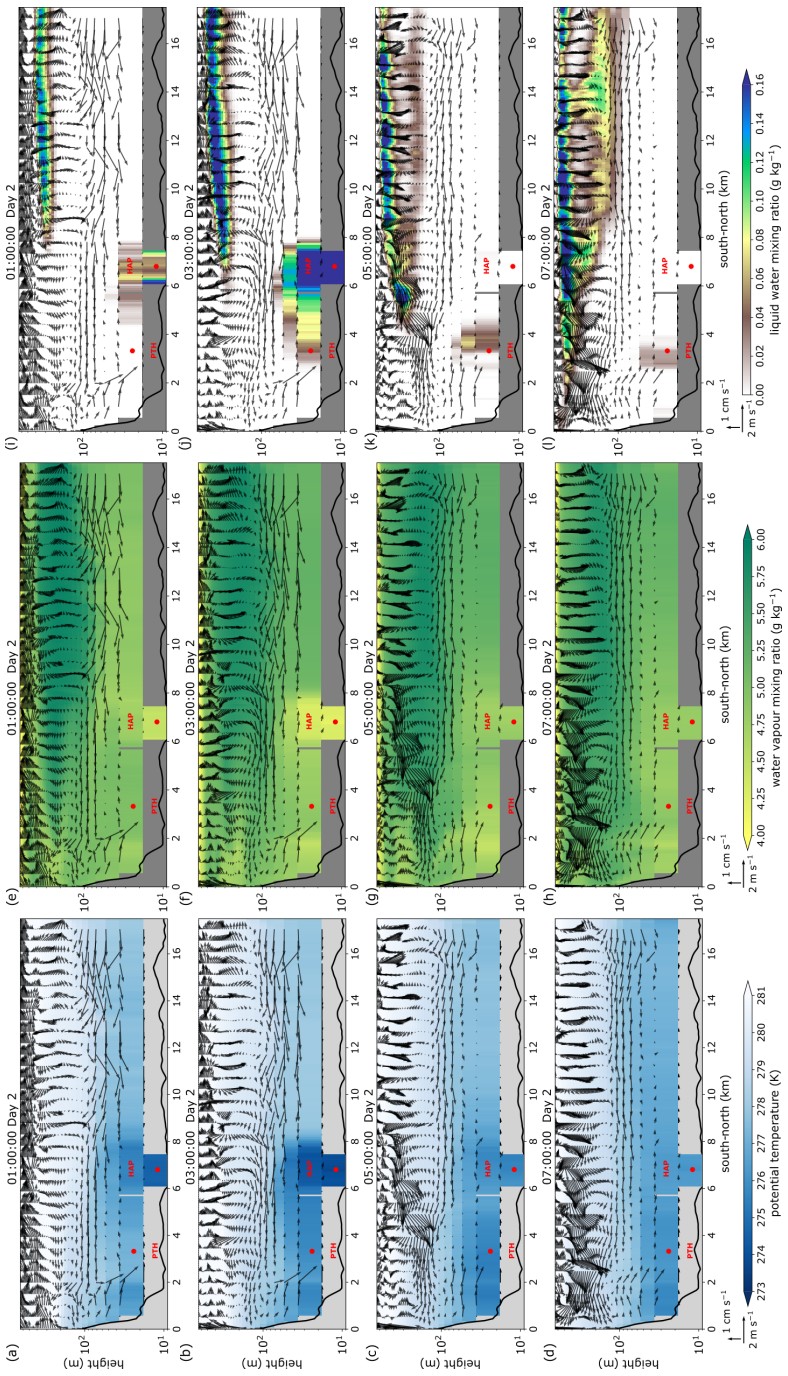

**Figure 6.** South–north vertical cross sections over central Hagley Park (west–east distance at approximately 12.2 km, see location marked in Figure 3a) showing wind vectors ($v$), potential temperature ($pt$) (a-d), water vapour mixing ratio ($qv$) (e-h), and liquid water mixing ratio ($ql$) (i-l). The solid back line at the bottom of each panel is the topography height based on the geospatial data input. Grey areas indicate the topography structure simulated by PALM. Note that a logarithmic scale is used on the y axis. The red dots in each panel indicate the relative south–north locations of PTH and HAP.

During the entire simulation, fog rarely formed over the northern half of the domain, and as shown in Figures 5a-c, the MRi values were near-zero or negative over this area. This is likely caused by the layer of low stratus that formed over this area at around 400 m above the ground, while the southern part of the simulation domain was under clear sky conditions (see Figure 6i). The formation of the low stratus layer is considered to result from high $qv$ at around 400 m at model initialisation (Figure 2c), with high values of $qv$ at approximately 400 m also visible in Figures 6e-h. The layer of low stratus absorbed the outgoing longwave radiation from the surface and re-radiated it to the surface over the northern section of D04, which is likely to have led to reduced surface cooling and subsequently a less stable near-surface layer over this area. Besides the low stratus clouds, the simulation featured clear sky conditions. The clouds and fog started to lift and dissipate around sunrise time (0736 LST), while patchy clouds were still presented 3 hours after sunrise. More investigation is required into the potential impacts of cloud. Since this is out of the main scope of this study, such impacts will not be discussed further.

## 5 Investigation of the impact of soil moisture heterogeneity

In this section, the soil moisture heterogeneity configuration is introduced. Although there is significant spatial heterogeneity across the domain presented in the baseline scenario, for efficiency we only explore the impact of soil moisture heterogeneity on fog for four selected sites (HAP, PTH, SWC, and WMR). The spatial heterogeneity of fog is first studied for these four sites followed by analysis of the impact of soil moisture heterogeneity using pseudo process diagrams and accumulated latent heat flux.

### 5.1 Soil moisture heterogeneity configuration

To include soil moisutre heterogeneity in the simulatoins, this study adopted the soil moisture index (SMI) calculation method for Landsat 8 imagery described by Avdan and Jovanovska (2016); Potić et al. (2017). SMI describes the proportion of actual soil water content relative to a known wilting point and field capacity (e.g., Zeng et al., 2004; Hunt et al., 2009). Our case study focuses on the spatial heterogeneity rather than accurate values of soil moisture. The derived SMI was therefore used directly as soil moisture here. The spatially heterogeneously distributed soil moisture of the simulation domain was first adjusted to have the same mean value as HOM. The soil moisture was then adjusted to amplify the signal of soil moisture heterogeneity. Finally, the soil moisture at each grid point was adjusted again such that the mean value of the entire domain is identical to HOM. The second adjustment (hereafter readjustment) was designed to amplify and help us understand the impact of soil moisture heterogeneity on fog development. Both the aforementioned adjustments were applied first to surface soil moisture, and then to soil moisture at all soil vertical levels by adjusting the value at each grid point such that the slope of the profile shown in Figure 2 did not change. Details of the adjustment are explained further in Appendix B.

The readjustment method was applied to an N-by-N readjustment area, where N is a selected number of grid points, with N applied in both west-east ($x$) and south-north ($y$) directions. For each N-by-N readjustment area, the mean soil moisture ($SM_{mean}$) was calculated. Then, if $SM_{mean}$ was greater than or equal to the soil moisture value for the homogeneous simulation ($SM_{HOM}$), the soil moisture did not change for the grid points inside the N-by-N readjustment area with soil moisture

greater than and equal to $SM_{mean}$. For grid points inside the N-by-N readjustment area with soil moisture less than $SM_{mean}$, the soil moisture of the grid points was increased to $SM_{mean}$. On the other hand, if $SM_{mean}$ was less than $SM_{HOM}$, the soil moisture value did not change for the grid points inside the N-by-N readjustment area with soil moisture less than $SM_{mean}$. For grid points inside the N-by-N readjustment area with soil moisture greater than $SM_{mean}$, the soil moisture of the grid points was decreased to $SM_{mean}$. An example of this soil moisture readjustment is shown in Figure B1. Following this readjustment, an N-by-N area with a higher mean value became wetter, while an area with a lower mean value became drier, although the internal heterogeneity of each N-by-N area was preserved.

In this study, the following numbers of west-east and south-north grid points were chosen in order to apply the readjustment method: 12, 18, 36, 54, and 108 (hereafter denoted as HET12p, HET18p, HET36p, HET54p, and HET108p, respectively). These numbers of grid points were chosen as the total number of grid points in each direction (216) of the D04 domain is divisible by them. The soil moisture content of the first soil layer of HET (no readjustment applied), and the soil moisture difference between HOM and all the heterogeneous simulations are shown in Figure 7. The soil moisture difference was calculated by subtracting the soil moisture of HOM from that of the heterogeneous simulation. This readjustment method amplifies the dry and wet signal from the original soil moisture heterogeneity while keeping the magnitude of soil moisture changes within a realistic range. Furthermore, to simulate the effects of the lowest and highest soil moisture availability, we conducted two extra simulations with halved and doubled soil moisture of HOM (hereafter Half and Double, respectively).

## 5.2 Results

### 5.2.1 Changes in fog duration and occurrence

In the environmental conditions of heterogeneous topography, land use, and soil moisture, the simulated formation and dissipation times of fog at each grid point are different in each simulation. Therefore, we focus on surface fog duration and occurrence to assess the sensitivity of fog to changes in soil moisture heterogeneity. Fog duration at the first model level was calculated for all the nine simulations and the results are compared in Figure 8. Figure 9 presents the difference in the locations of fog occurrence locations between HOM and the other simulations. The comparison shows significant spatial variation in fog duration between HOM and the other simulations. However, the strong topographical and meteorological steering restricts deviations in the simulated locations of fog occurrence under this typical meteorological scenario. Double and Half have led to an increase and decrease in the areas of fog occurrence, respectively (Figure 9a and 9h), although the changes in fog location for the heterogeneous simulations are not significant as shown in Figure 9. While Double and Half represent the highest changes in the total soil moisture content, the changes in fog duration and locations of occurrence are not consistent for the two simulations. In Double, fog duration increased near WMR, HAP, and PTH, but decreased around SWC, compared to HOM. The increase in soil moisture of Double led to more fog occurring to the south of WMR, while there is no significant change for the rest of the domain. For Half, fog duration decreased around HAP, PTH, and SWC, while the duration difference is low around WMR. As a result of water content being removed from soil, the area of fog occurrence is smaller in Half compared to HOM.

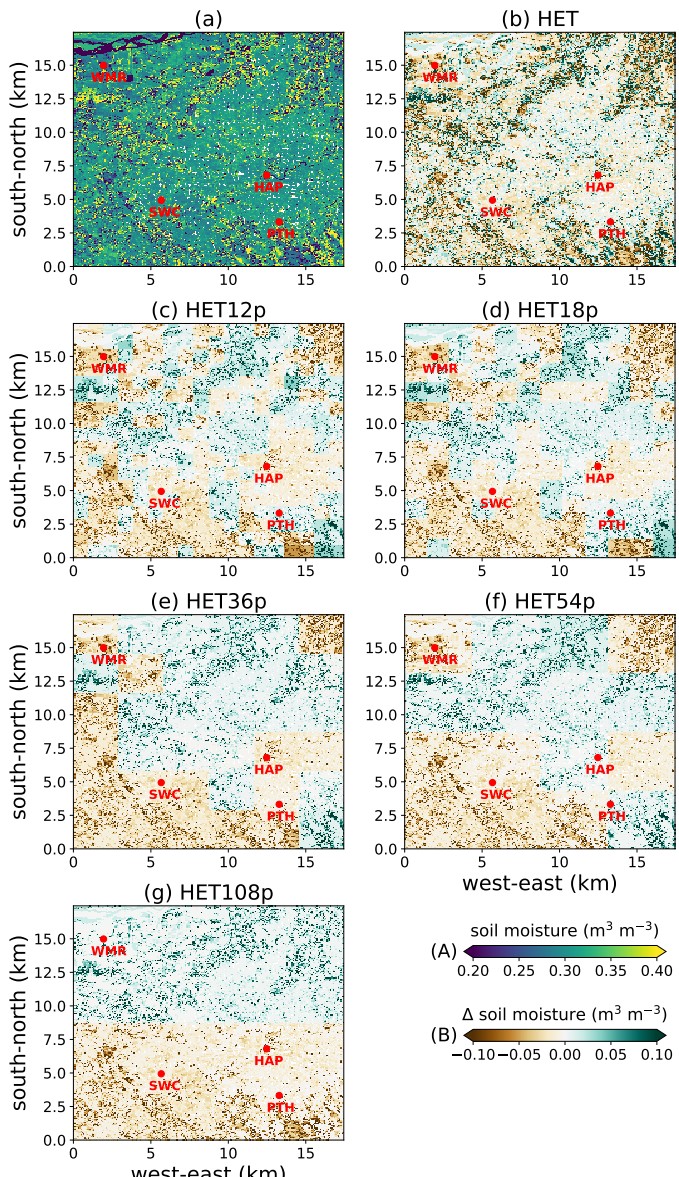

**Figure 7.** Soil moisture for D04 in HET (a), and soil moisture difference between the heterogeneous simulations and the homogeneous simulation (HOM) for (b) HET, (c) HET12p, (d) HET18p, (e) HET36p, (f) HET54p, and (g) HET108p. Colour legend (A) applies to panel (a) only. Colour legend (B) applies to panels (b)-(g). The difference was a result of the soil moisture in the homogeneous simulation subtracted from that of the heterogeneous simulation. Green areas in panels (b)-(g) indicate that soil moisture in the heterogeneous simulation is higher than in HOM, while brown areas indicate drier soil in the heterogeneous simulation. The four locations of interest are marked by red dots.

In this particular fog case study, the amplified soil moisture heterogeneity does not significantly alter the fog occurrence patterns. Nevertheless, fog duration shows high sensitivity to soil moisture heterogeneity and the resulting changes in fog

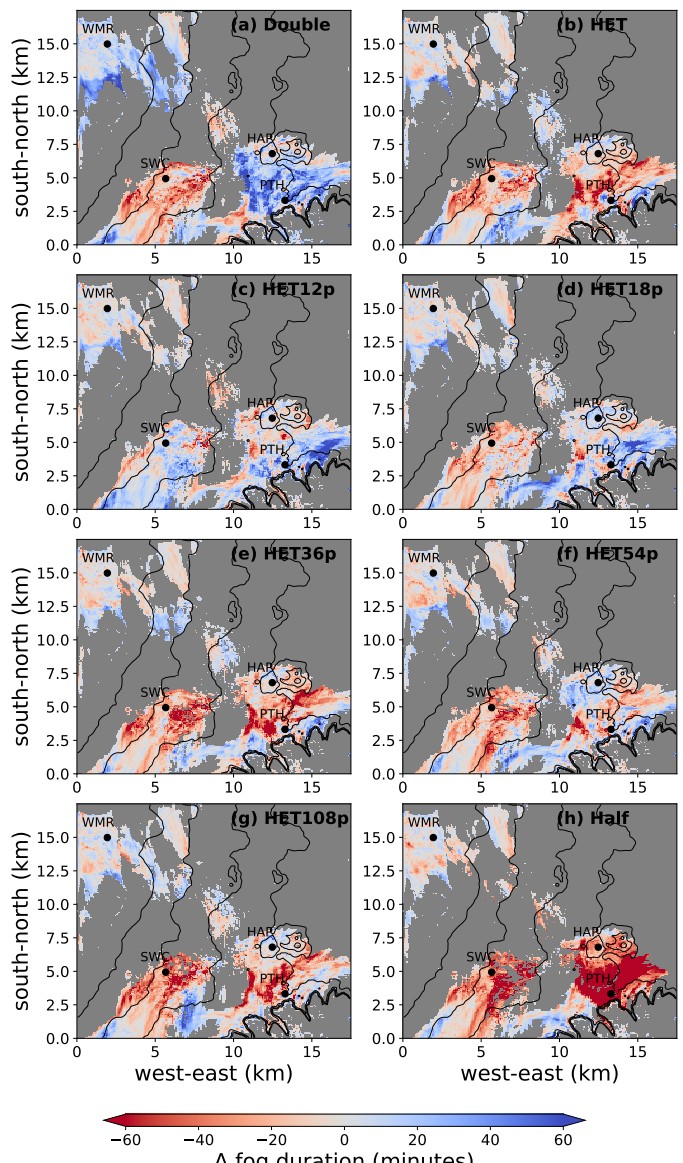

**Figure 8.** Fog duration difference in minutes between each simulation and HOM for Double (a), HET (b), HET12p (c), HET18p (d), HET36p (e), HET54p (f), HET108p (g), and Half (h), respectively. The fog duration difference is the result of the heterogeneous simulations subtracted from HOM. Red indicates longer fog duration in HOM and blue indicates the opposite. Grey areas are where fog did not occur in both HOM and the corresponding simulation. The four locations of interest are marked by black dots. Dark grey contour lines in each panel indicate terrain height labelled in Figure 3a.

duration can be more than 50 minutes for some areas (Figures 8b-g). However, there is no direct evidence to link the changes in soil moisture to the changes in fog duration and occurence locations. When compared at each grid point, the changes in fog

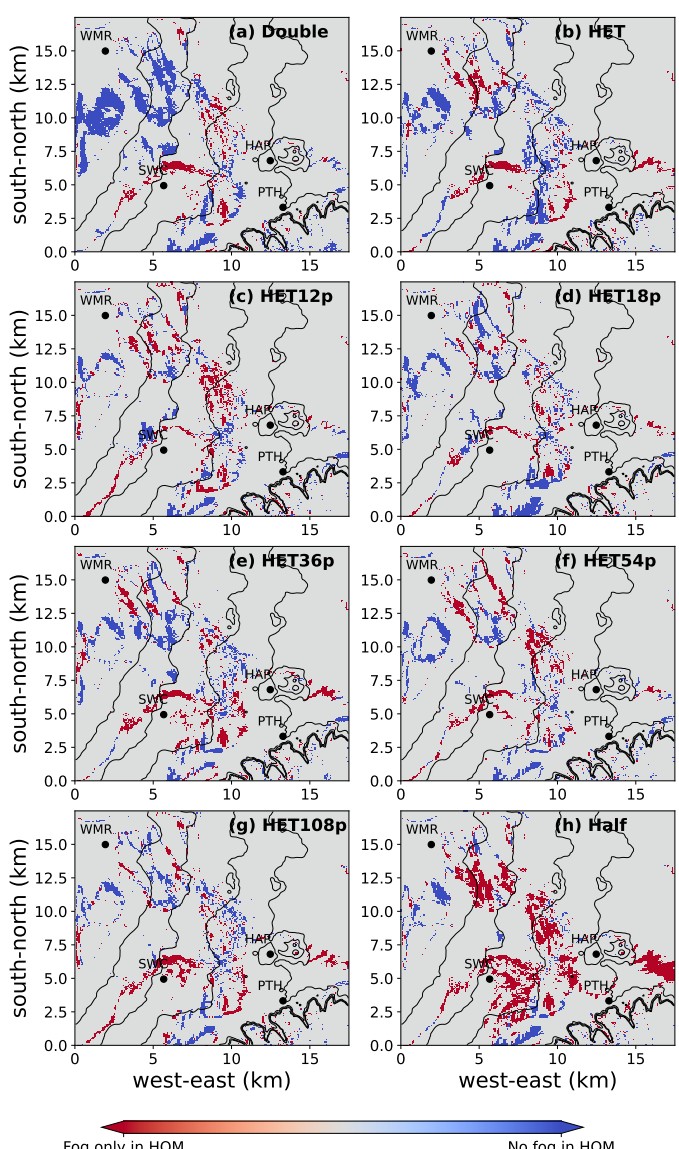

**Figure 9.** Similar to Figure 8, but for difference in locations of fog occurrence between the other simulations and HOM. Red indicates locations where fog only occurred in HOM and blue indicates locations where fog only occurred in the corresponding simulation of each panel.

duration and occurrences shown in Figures 8 and 9 do not mirror the changes in soil moisture shown in Figure 7. For example, HET and HET108p have significantly different soil moisture patterns (Figures 7b and 7g), while the decrease or increase in fog duration (Figures 8b and 8g) does not show considerable variation between the two simulations. Fog occurrence locations are also similar in the two simulations (Figure 9b and 9g). However, both HET12p and HET18p show areas of increase in fog

**Table 4.** Fog formation and dissipation time marks for HAP, PTH, SWC, and WMR at 1 km scale for all fog simulations.

| | HAP | | PTH | | SWC | | WMR | |
|---|---|---|---|---|---|---|---|---|
| | Formation | Dissipation | Formation | Dissipation | Formation | Dissipation | Formation | Dissipation |
| HOM | 0145 LST | 0509 LST | 0340 LST | 0824 LST | 0713 LST | 0828 LST | 0741 LST | 0934 LST |
| Half | +5 | -29 | +115 | +7 | +10 | -17 | -1 | +5 |
| Double | +6 | +16 | -10 | +2 | +13 | +1 | +1 | +7 |
| HET | +3 | -1 | -10 | +19 | +17 | -4 | +0 | +2 |
| HET12p | +4 | +13 | +0 | +14 | -1 | +5 | +2 | +1 |
| HET18p | +0 | +11 | -36 | -10 | -1 | +1 | -1 | +1 |
| HET36p | -2 | +6 | +15 | +9 | +33 | +3 | -1 | +4 |
| HET54p | -1 | +25 | +1 | +11 | +14 | -10 | -2 | +3 |
| HET108p | -1 | +14 | -1 | +12 | +14 | -13 | +1 | -3 |

duration near PTH in the southeast of D04 (Figures 8c and 8d), while this pattern is absent in other heterogeneous simulations. The HET simulation results in a decrease in fog duration over the area near SWC (Figure 8b), while HET12p shows an increase in fog duration in this area (Figure 8c). This pattern was not present in HET18p and HET36p (Figures 8d and 8e), where the heterogeneous signals were further amplified.

### 5.2.2 Spatial heterogeneity of fog

Due to high spatial heterogeneity in the simulated fog, point-by-point comparison does not show a clear correlation between fog duration and soil moisture heterogeneity. Therefore, here we focus on the simulation results at 1 km and 3 km scales at the four selected locations (HAP, PTH, SWC, and WMR). First, similar to Bergot and Lestringant (2019), the spatial coverage of fog was calculated in the squares centred at the sites of interest with sides of 1 km and 3 km (Figure 10). For HAP and WMR in HOM, the fog coverage frequency at 1 km scale is similar to the fog occurrence at the grid point (Figure 10a and 10j), indicating a relatively homogeneous fog layer. At PTH, the fog coverage at 1 km is patchier with fog coverage only reaching 95% approximately 2 hours after fog was first detected (around 0200 LST; Figure 10d). At SWC, the fog detected around 0400LST is patchy and the fog coverage at both 1 km and 3 km scales was below 20%. We therefore only focus on the fog event at SWC between 0600 LST and 0900 LST in the following discussion. At 3 km scale, the maximum fog coverage barely exceeded 80% at all the four sites suggesting significant spatial heterogeneity of fog occurrence throughout the entire fog life cycle for all four sites.

Table 4 shows the sensitivity of fog formation and dissipation times at 1 km scale. Here, based on Bergot and Lestringant (2019), the fog formation phase is recognised as the time when fog coverage is between 5% and 95%. We recognise the dissipation phase as when fog coverage falls from 95% to 5%. Regarding HAP, no significant variation can be seen during the fog formation phase. The formation and dissipation times for HET are similar to HOM, while the differences in dissipation time are greater in other heterogeneous simulations. Halved soil moisture coincided with an earlier dissipation of 29 minutes,

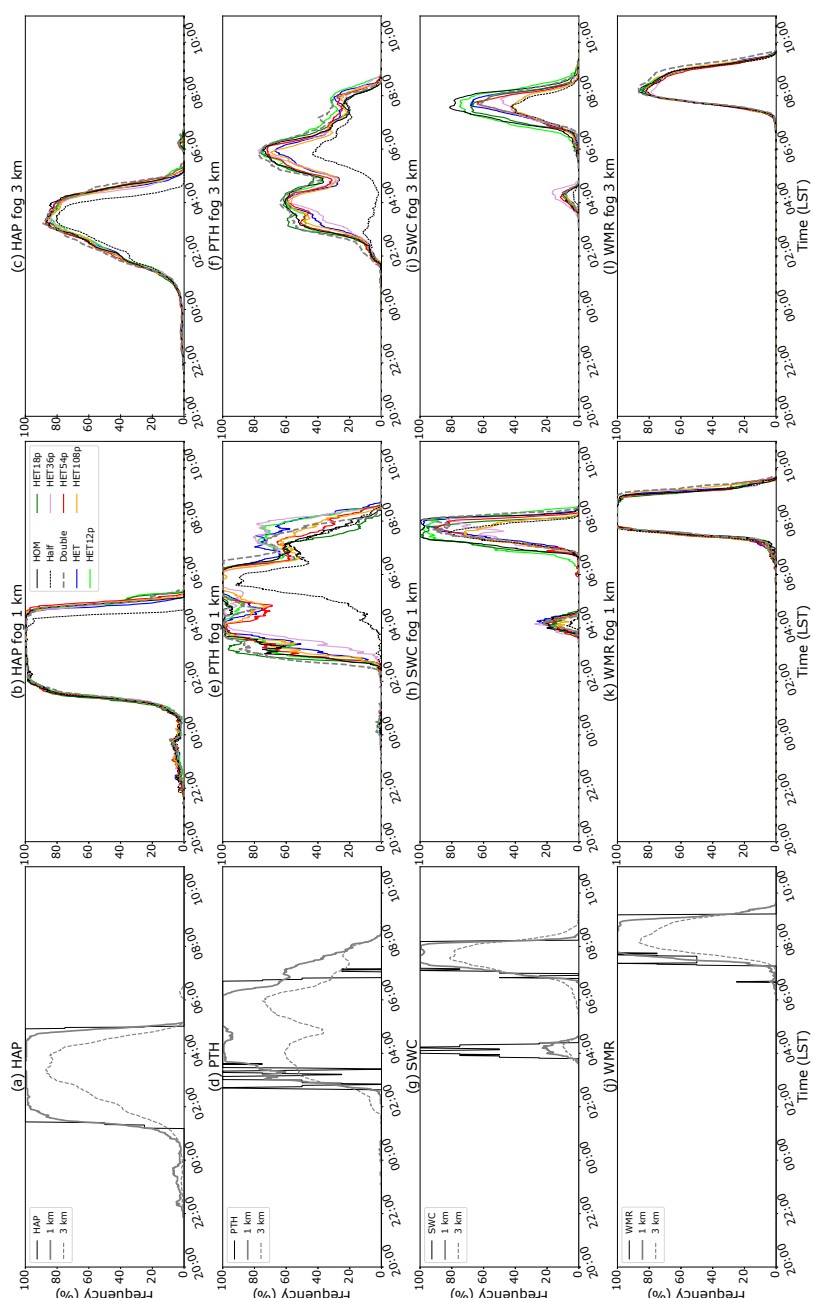

**Figure 10.** Time series of fog coverage for HOM (a, d, g, and j), and for all simulations over a 1 km by 1 km square (b, e, h, and k), and a 3 km by 3 km square (c, f, I, and l) for HAP, PTH, SWC, and WMR. In panels (a), (d), (g), and (j), the thin black line indicates fog occurrence at the specific site. Colours refer to panel (b) legend.

while doubled soil moisture presents a late dissipation by 16 minutes. Nevertheless, the changes in dissipation time do not show direct correlation with the local soil moisture content in the heterogeneous simulations. The 1 km square around HAP is associated with a lower soil moisture in HET108p (Figure 7h), but HET108p still showed a later dissipation time compared to HOM.

For PTH, at 1 km scale, the formation phase of fog is significantly delayed by almost 2 hours in Half, while the changes in Double are less noticeable. The changes in formation and dissipation times do not show consistency with respect to changes in soil moisture heterogeneity. The differences between each heterogeneous simulation can be over 50 minutes. HET18p presents a delayed fog formation by 36 minutes, while fog formation was 15 minutes earlier in HET36p. Similar to HAP, lower soil moisture at PTH in HET108p does not coincide with less fog duration as shown in Half.

At SWC, soil moisture heterogeneity led to significant spatial heterogeneity of fog at both 1 km and 3 km scales. At 1 km scale, HET36p, HET54p, and HET108p did not reach 90% during the entire fog period. For comparison purposes, the formation phase is hence defined as the period when fog coverage is between 5% and 70% at SWC. The changes in soil moisture and its heterogeneity generally led to delay in fog formation, while the dissipation time varies within a 20 minute range. At 3 km scale, compared to HOM, the maxima of the spatial fog coverage decreased in all other simulations, with Half and HET108p at almost 50%. Doubled soil moisture did not lead to higher fog coverage at both 1 km and 3 km scales at SWC.

The variation at WMR between all simulations is the least significant regardless of the scale. The changes in formation and dissipation times are within 10 minutes. This is in agreement with the suggestion that the fog event at WMR is related to evaporation and advection near the Waimakariri River as described in Section 4. These results at the four sites indicate that local soil is not the only source of moisture availability in the atmosphere and advection of air parcels may play a more important role regarding the transportation of water vapour. At the four sites, the dominant processes driving the fog are different. Therefore, no consistency across the four sites can be found regarding the impact of soil moisture and its spatial heterogeneity.

### 5.2.3 Driving processes of fog heterogeneity

To obtain greater insights into the role of soil moisture heterogeneity, we investigate the dominant processes driving the fog at the four selected sites. We have adopted the pseudo-process diagram used in Steeneveld and de Bode (2018) and Bergot and Lestringant (2019). Similar to Bergot and Lestringant (2019), pseudo-process diagrams of surface net radiation ($Rnet$) versus vertical temperature gradient ($dT/dz$) were created for all grid points within the 1 km and 3 km scales for HAP, PTH, SWC, and WMR (Figure 11). Since we only focus on fog that occurred near the surface, $dT/dz$ was only derived using surface temperature and temperature at the lowest model level. The goal of these diagrams is to identify whether fog formed and developed locally, or whether it was a result of advection. Typical radiation fog is associated with strongly negative Rnet and a temperature inversion ($dT/dz>0$) at formation, as well as nearly constant stability and a slow increase in $Rnet$ while the near-surface layer becomes saturated and the fog layer grows vertically. When advection processes are involved, the near-surface layer is weakly stable or unstable ($dT/dz <0$) associated with a rapid increase in $Rnet$.

At HAP, significant variability exists in Figure 11 at both 1 km and 3 km scales. This is due to heterogeneity in land use, topography, and spatial heterogeneity of the fog layer itself. This variability is more pronounced at the 3 km scale (Figure

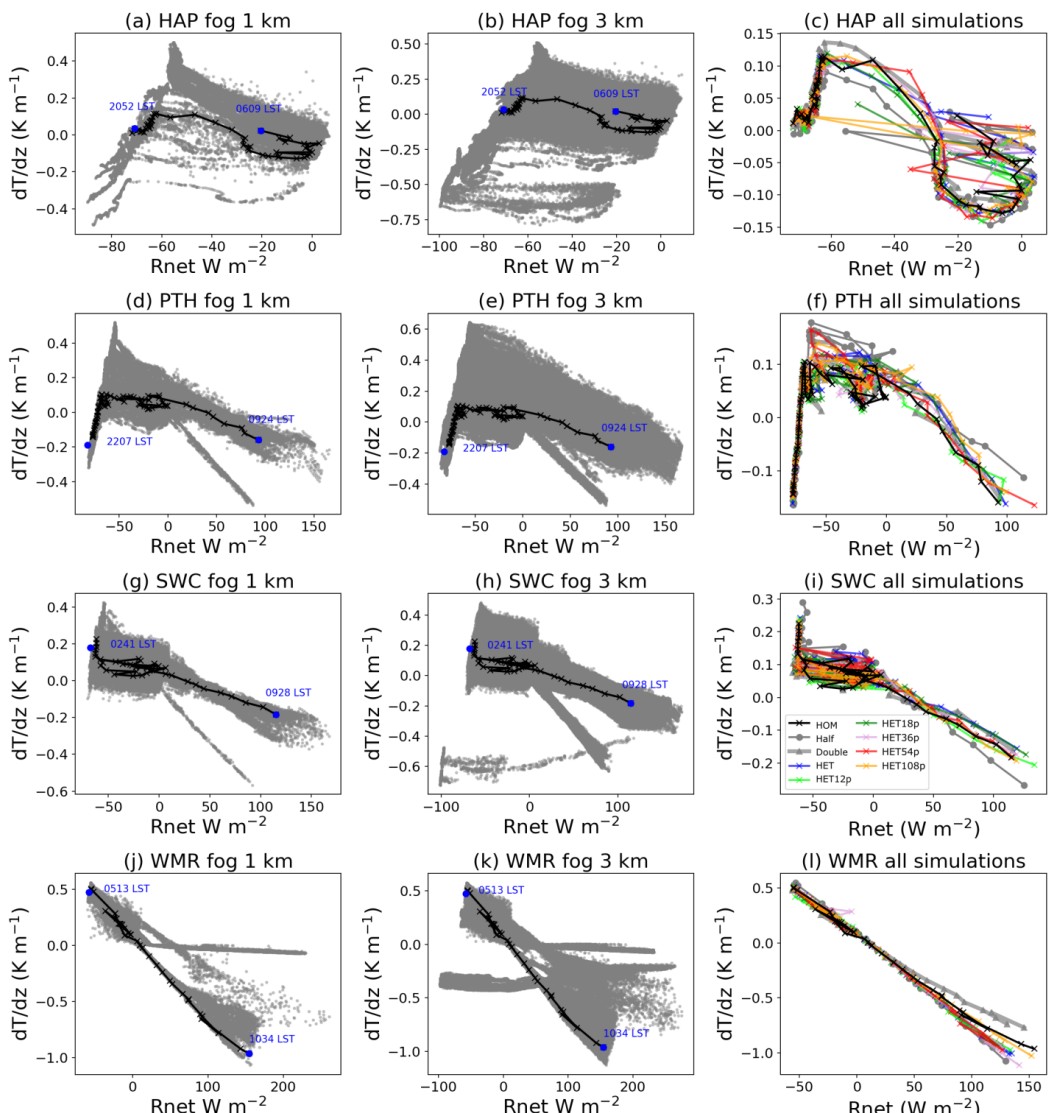

**Figure 11.** Pseudo diagrams between surface stability (dT/dz) and net radiation (Rnet) for the four selected sites: HAP (a, b, c), PTH (d, e, f), SWC (g, h, i), and WMR (j, k, l). The first column (a, d, g, j) shows diagrams at 1 km scale for HOM, the second column (b, e, h, k) shows diagrams at 3 km scale for HOM, and the third column (c, f, i, l) shows the trajectories for all simulations. The trajectory for the selected site is marked by the solid line. In the first two columns, the grey dots are obtained from all grid points at 1 km and 3 km scales, respectively. Colours for panels (c), (f), (i), and (l) refer to panel (i) legend. Data point frequency is every 10 minutes.

11b). The first phase of the trajectory shows an increase in stability associated with an increase in $Rnet$ suggesting that the formation of this fog event is associated with cooling in a stable near-surface layer. Saturation is reached when $dT/dz$ becomes

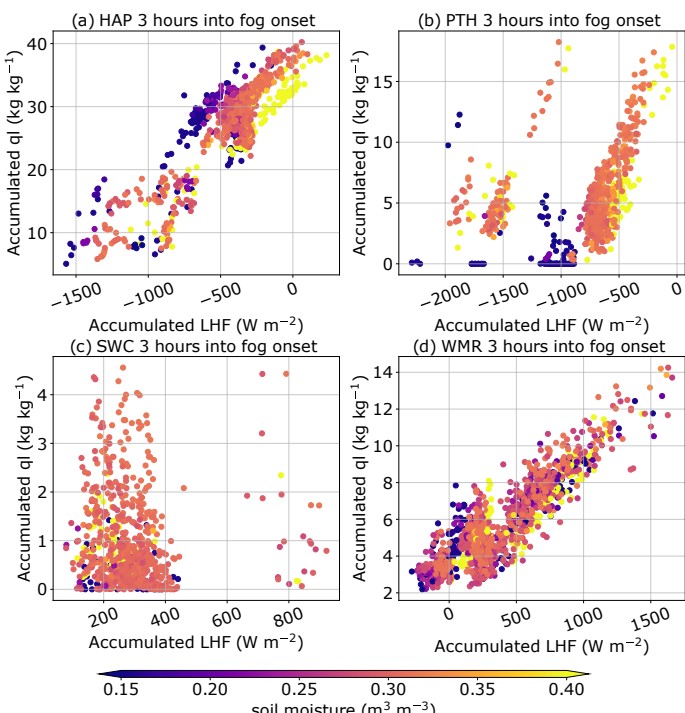

**Figure 12.** Scatter plots showing accumulated latent heat flux (LHF) against accumulated $ql$ coloured by soil moisture for (a) HAP, (b) PTH, (c) SWC, and (d) WMR. Data samples were obtained from each grid point at 1 km scale for all simulations. The accumulation period is 3 hours into fog onset.

near constant with varying Rnet, after which no correlation is presented between $dT/dz$ and $Rnet$. This is likely caused by the increase of turbulence mixing, when fog dissipated at HAP around 0430 LST as described in Section 4.

At PTH, the pseudo diagrams present less variability of the processes involved. The first phase is similar to HAP such that both Rnet and the stability were increasing. This fog, however, did not develop into a deep adiabatic fog layer. Bergot and Lestringant (2019) showed that $Rnet$ and $dT/dz$ are close to zero when a fog layer becomes deep adiabatic. In this case, the near-surface layer became unstable and $Rnet$ exceeded zero at around sunrise time (0734 LST) and fog started to dissipate.

At both SWC and WMR, the trajectories started with weakly negative $Rnet$ and a stable near-surface layer. The evolution of

the trajectories showed an increase in $Rnet$ associated with a decrease in stability, implying that the fog may not be a radiation fog and is more likely to be advection of a fog layer. As this study mainly focuses on radiation fog, the processes involved are not further discussed.

Comparison between all simulations for the four selected sites shows similar patterns in the pseudo diagrams (Figure 11c, f, i, and j). At HAP and PTH, the first phase (an increase in both $Rnet$ and $dT/dz$) is almost identical for all simulations.

However, the saturation phase ($dT/dz$ nearly constant) shows significant variability across the simulations for HAP and PTH.

Cross-simulation variability is low for SWC and WMR confirming that advection processes are the primary drivers for the corresponding fog events.

To further analyse the impact of soil moisture heterogeneity, Figure 12 relates accumulated latent heat flux ($LHF_{Acc}$) to accumulated liquid water mixing ratio ($ql_{Acc}$) during the period of 3 hours into fog formation for the four selected sites at 1 km scale. Data points in Figure 12 were obtained from all simulations. $LHF_{Acc}$ was calculated for the 3 hour period before sunset, the period between sunset and fog formation, and the 3 hour period into fog formation (shown in Figure B2). The $LHF_{Acc}$ for the former two periods do not show a correlation with the $ql_{Acc}$ for the 3 hours into fog formation (not shown). Therefore, here we only focus on the 3-hour period into fog onset, which aligns with the saturation phase at HAP and PTH shown in the pseudo diagrams (Figure 11).

As shown in Figure 12a, at HAP, higher $LHF_{Acc}$ is generally associated with higher $ql_{Acc}$. In addition, although higher soil moisture shows a positive correlation with $LHF_{Acc}$ for fixed values of $ql_{Acc}$, an increase in soil moisture does not coincide with a higher $ql_{Acc}$. At PTH, the relationship between $LHF_{Acc}$ and $ql_{Acc}$ is not linear, while a positive correlation can be found for $LHF_{Acc}$ ranging between -2000 and -1500 W m$^{-2}$ and between -1000 and 0 W m$^{-2}$ (Figure 12b). This may be due to variability in land use and vegetation type over the Porth Hills area (Figure 1d). At HAP, the main vegetation type includes bare soil, short grass, and deciduous needleleaf trees. At PTH, in addition to short grass and bare soil, the land surface includes mixed farming, tall grass, and deciduous broadleaf trees. The topography at PTH is also more complex compared to HAP (Figure 3a). Therefore, we consider $LHF_{Acc}$ generally correlates positively with $ql_{Acc}$ at PTH, but different sectors are presented. These profiles agree with the conclusion that localised cooling and the consequent condensation of water vapour are the main drivers at HAP and PTH. However, similar to HAP, no correlation was presented between soil moisture and $ql_{Acc}$, although high soil moisture is associated with higher $LHF_{Acc}$.

At SWC, all the three variables do not show any relationship, while at WMR an increase in $LHF_{Acc}$ relates to an increase in $ql_{Acc}$, Both sites do not show a significant impact of soil moisture. The $LHF_{Acc}$ is positive for the two sites, which aligns with the conclusion that both fog events at SWC and WMR are caused by advection processes. The difference between Figure 12c and Figure 12d implies that the primary processes involved in the two fog events are not the same. In addition, the changes in soil moisture impact the fog events at SWC and WMR differently. When soil moisture is doubled, the area close to WMR coincides with an increase in both fog duration and fog occurrence (Figure 8a and Figure 9a). This is the opposite for SWC as its surrounding area shows a decrease in fog duration in the Double simulation, while a slight decrease appears in the area of fog occurrence (8a and Figure 9a).

## 5.3 Discussion

These numerical simulations show significant spatial heterogeneity of fog occurrence. The processes involved in fog formatoin can be considerably different in just one case study, in agreement with the fog events simulated by Bari et al. (2015) and Bergot and Lestringant (2019). Our results show that soil moisture and its heterogeneity can lead to changes in fog occurrence as well as the timing of fog formation and dissipation. In Maronga and Bosveld (2017), tests of radiation fog sensitivity to the initial state of soil moisture suggested that changes in soil moisture content do not affect fog formation time, although fog

lifting time is affected. Our simulations do not agree with their findings, as both the formation and dissipation times vary in each simulation. As both Maronga and Bosveld (2017) and our study used PALM, we believe the difference in the results is due to a different simulation setup. For example, spin-up time for Maronga and Bosveld (2017) was only 35 minutes, while our spin-up time was 24 hours, so that daytime evaporation from the soil is not included in their simulations. The fog lifting and dissipation period in Maronga and Bosveld (2017) occurred during sunrise, while fog dissipation time in our study varied

across the simulation domain. Maronga and Bosveld (2017) simulated their radiation fog event over flat terrain with land use type configured as grassland only and did not include spatial heterogeneity in soil moisture. The fog simulated in their study therefore does not present significant spatial heterogeneity. In this study, we conducted two simulations with halved and doubled soil moisture. Nevertheless, the changes in fog occurrence, formation time, and dissipation time do not show a linear correlation with changes in soil moisture across the domain. At HAP and PTH, where fog reflected more localised processes,

fog duration decreased by more than 20 minutes when soil moisture was halved. However, this relationship is less clear for SWC and WMR. It should be noted that this study uses a coarse grid spacing (horizontal grid spacing of 81 m and vertical grid spacing of 18 m) compared to Maronga and Bosveld (2017) (grid spacing finer than 4 m). Our simulations did not resolve large eddies, and hence, the turbulence transport could be expected to differ significantly from Maronga and Bosveld (2017). The grid spacing used in this study falls in the "Grey Zone" of turbulence (e.g., Honnert et al., 2020), which allows us to carry

out multiple experiments relatively quickly over a large area, although the representation of the turbulence is less accurate. To provide more confidence in our simulations, we have carried out sensitivity experiments on the vertical grid spacing of D04 (from 18 m to 9 m and 6 m). Our conclusions remain unchanged when a finer vertical grid spacing is used (see supplements). In addition, running simulations over an area of approximately 17.5 km × 17.5 km with grid spacing finer than 4 m is not computationally feasible. Future work should therefore be carried out using a finer grid spacing when suitable computation

resources become available.

As a range of processes are involved and the dominant processes vary spatially, the impact of soil moisture and its heterogeneity is reduced. The pseudo-process diagrams show that fog is not homogeneous at both 1 km and 3 km scales for most of the focused areas. Such spatial heterogeneity of fog and the processes involved makes identifying and quantifying the impact of soil moisture heterogeneity difficult. Analysis of $LHF_{Acc}$ does not show direct correlation with soil moisture implying

advection of water vapour in the atmosphere may play a more important role compared to evaporation from the soil. However, the heterogeneity of soil moisture is still considered important. As shown in Figure 7, over 70% of the grid points in the heterogeneous simulations are associated with a a range of values less than 0.05 $m^3$ $m^{-3}$. It should be noted that this value falls within a typical range of soil moisture values over a diurnal cycle (between -0.1 $m^3$ $m^{-3}$ and 0.1 $m^3$ $m^{-3}$; Meng and Quiring, 2008). Furthermore, the mean values of soil moisture for D04 for all simulations are identical. The differences in fog

formation and/or dissipation timing between each heterogeneous simulation can be more than 50 minutes for some areas. This highlights the importance of including soil moisture heterogeneity in fog forecasting, as fog duration can be very sensitive to small changes in soil moisture and soil moisture heterogeneity.

## 6 Conclusions

A set of numerical fog simulations was conducted to investigate the impacts of meteorological controls on radiation fog in
Christchurch using the microscale model PALM. As Christchurch's observational network does not have a great spatial and
vertical coverage, we did not attempt to replicate a real radiation fog event in our simulations. Rather, we conducted semi-
idealised simulations using profiles from a selected radiation fog event to create a radiation fog scenario as guidance for future
fog research and forecasting. We aimed to understand the impact of soil moisture heterogeneity on radiation fog, with our case
study showing that both mesoscale and microscale dynamics are important. The meteorological controls in a heterogeneous
environment were illustrated. To the best of our knowledge, this study is the first to include heterogeneous topography, land
use, and soil moisture in radiation fog simulation at the microscale.

Overall, the macrostructure of fog occurrence and distribution is highly controlled by topography and the mesoscale mete-
orology, while fog duration is sensitive to changes in soil moisture heterogeneity at the microscale. As illustrated in Section
4, the synoptic situation, drainage flow, local topography, the occurrence of overlying clouds, and local radiative cooling all
affect the onset, duration, dissipation and spatial distribution of fog. At the microscale, the readjustment of soil moisture het-
erogeneity does not alter the general near-surface wind flow structure and the location of fog occurrence in the simulations.
While the range of soil moisture heterogeneity values applied in this case study is relatively small, all the six heterogeneous
simulations resulted in noticeable variations of fog formation, dissipation, and subsequently duration. For some areas, when
compared to the homogeneous setting, the fog duration can vary by more than 50 minutes, which could be significant for
aviation forecasts. The spatial variation in fog duration, however, does not mirror the spatial heterogeneity in soil moisture.
Even when soil moisture was doubled and halved, the changes of fog are not consistent across the simulation domain. The fog
presented in the simulations is spatially heterogeneous. Four sites were selected to explore the driving processes of fog and
the impact of soil moisture and its heterogeneity. With the help of pseudo-process diagrams, we identified that fog types vary
across the simulation domain. For radiation fog, increase of $LHF_{Acc}$ during the 3-hour period into fog formation can lead to a
denser near-surface fog layer. Quantifying the impact of soil moisture and its heterogeneity, however, requires further research
through, for example thorough additional case studies, as only one case study is presented here.

Lack of observational data is one of the biggest challenges. More observational data from, for example, field campaigns,
are necessary to advance fog research. Development of an accurate soil moisture observation network may be necessary, in
order to apply heterogeneous soil moisture derived from satellite observations for operational fog forecasts. The soil moisture
index values were directly used in this study to investigate the impact of soil moisture heterogeneity on radiation fog and do
not represent the true soil moisture for a real case. Due to the difficulty in spatial analysis and the significant computational
cost, we only carried out simulations at a horizontal grid spacing of 81 m. In our high-resolution mesoscale simulations, most
of the eddies are not resolved. With PALM's high scalability at microscale, the grid spacing of the simulations should typically
be finer so that turbulence structures can be better resolved and captured. For the purpose of fog forecasting, one may aim at
a vertical grid spacing less than 2 m near the surface (e.g., Tardif, 2007) with accurate soil moisture derived from Landsat 8
observations. With finer grid spacing, idealised simulations with homogeneous land use or topography should be carried out

along with observational data. Singling out the components of land surface heterogeneity may provide greater insight into the impact of soil moisture heterogeneity on the fog life cycle. Furthermore, our simulations only used the RRTMG scheme in PALM for computation of radiation and did not include the three-dimensional RTM. As discussed in Salim et al. (2022), choices of the radiative transfer processes included in PALM can change the flow field considerably in an urban environment. The RTM applied in PALM neglects the absorption, scattering, and thermal emission by air masses (Krč et al., 2021; Salim et al., 2022), and hence its application is limited in case of fog simulations. A recent ongoing development regarding three-dimensional radiative transfer in PALM is the implementation and integration of the TenStream radiative transfer model. TenStream is capable to consider the effects of three-dimensional radiative transfer on the atmospheric heating rates or dynamic heterogeneities such as moving clouds or fog (Jakub and Mayer, 2015, 2016). With such development and implementation of the radiation model in PALM, TenStream should be considered and utilised for future fog simulations. Due to the inclusion of the plant canopy, only a one-moment microphysical scheme was used in this study. Although, as described in Schwenkel and Maronga (2019), the use of a one-moment scheme does not affect the general structure of fog life cycle, future work may aim to apply two-moment microphysical schemes for a more realistic representation of the microphysics. Identifying the contribution from soil moisture to fog was not achieved by analysis of accumulated latent heat flux. One possible way to further identify the impact of soil moisture heterogeneity is using a Lagrangian method to identify the sources of heat and water vapour. For example, a Lagrangian method was applied by Dale et al. (2020) to identify heat sources near the Ross Sea Polynya. Furthermore, only cyclic boundary conditions were used in our simulations. Future fog studies should consider using non-cyclic boundary conditions from NWP models to conduct more realistic fog simulations. This can be achieved using the offline nesting feature embedded in PALM and tools such as WRF4PALM (Lin et al., 2021) and INIFOR (Kadasch et al., 2021).

In conclusion, with the inclusion of heterogeneous topography, land use, and soil moisture, the present simulations demonstrate significant complexity. Spatial analysis is difficult within such a heterogeneous environment. Multiple meteorological controls can modify the local airflow structure and consequently the spatial structure and duration of fog. The contribution of soil moisture heterogeneity appears to be less significant than suggested by previous studies, in which complex topography and/or land surface characteristics were not adequately represented. Nevertheless, our study highlighted that soil moisture heterogeneity can lead to changes in radiation fog duration even in a complex environment, where both land use and topography are heterogeneous.

*Code availability.* This study used the free and open-source WRF model system V4.2 and the WRF Preprocessing System (WPS) V4.2 (download is available at https://github.com/wrf-model/WRF/releases, last access: 26 October 2022). The PALM model system 6.0 used in this study is a free and open-source numerical atmospheric model. PALM source code is available online (http://palm-model.org, last access: 26 October 2022) under the GNU General Public License v3. The exact PALM model source code (revision 4829) is available at https://palm.muk.uni-hannover.de/trac/browser? rev=4829 (last access: 26 October 2022).

*Data availability.* All PALM input files for the heterogeneous case described in Section 3, including the RRTMG input files, the static driver,
the dynamic driver, and its configuration file for all the simulation domains, are available in the Supplement.

**Appendix A: Fog event observations and synoptic conditions**

The upper air measurements recorded at 0000 LST 5th August were used in this study. This particular set of data was chosen
as it was recorded during an in situ observational campaign which took place from June to October 2001. In this campaign,
an acoustic sodar was operated at CHA adjacent to an AWS which provides measurements of temperature, wind speed and
direction, visibility, cloud height, etc. (Osborne, 2002). These upper air measurements were available before fog onset, which
is suitable for model initialisation. The time series of AWS and sodar observations, and the synoptic conditions for this fog
event are shown in Figures A1 and A1. In addition, this particular fog event has characteristics that agree with those identified
in a recent Christchurch fog climatology study (Lin et al., 2023a). In Christchurch, winter (June, July, and August) has the
highest frequency of radiation fog occurrence. Near-surface temperature inversions represent a key characteristic of radiation
fog formation and development, which is typically associated with sodar observations of an inversion layer situated between
50 m and 100 m above the surface (Figure A1). Both sodar and AWS observations show low wind speeds near the surface
(Figure A1) during this event, indicating a calm and relatively stable near-surface layer. The ERA5 reanalysis of mean sea level
pressure (MSLP; Figure A2) shows anticyclonic conditions indicative of clear sky conditions. Data for this fog event were
therefore used to carry out simulations in this study.

The time series of the AWS and sodar observations for the selected fog event are shown in Figure A1. This radiation fog
event occurred during the sodar operational period in the winter of 2001. The first order difference of the vertical profiles of
the sodar backscatter is derived along the height axis to highlight the elevation of the temperature inversion (Figure A1a). The
high values indicate where the backscatter signal is the strongest, which is where the inversion occurred. Some strong signals
at the lowest levels (approximately 10 m) are due to ground clutter. At around 2200 LST on 5th August, the inversion started
to develop at around 50 m. The inversion further developed to around 120 m by 0500 LST 5th August and visibility decreased
to below 1000 m. The inversion was sustained over the next 10 hours and the visibility only increased to above 1000 m after
1200 LST 6th August. The AWS observations at 10 m above ground show westerly and north-westerly winds near the surface
before the fog formed, indicating that the air mass was moving down the Canterbury Plains towards the coast, while the sodar
observations show northerly and easterly winds at 30 m. The surface wind speed was below $1.5\ \mathrm{m\,s^{-1}}$ between 1800 LST 5th
August and 1300 LST 6th August, while the wind speed at 30 m was stronger, but in general below $3.5\ \mathrm{m\,s^{-1}}$. Significant
cooling was recorded between 1800 LST on 5th August and 1300 LST on 6th August (from approximately 7 °C to –1 °C), and
relative humidity was above 97%. Observations from the AWS show that the cloud height was above 7500 m before fog onset,
but no cloud base lowering was recorded. The cloud base was low after 1300 LST 6th August because the fog layer developed
into mist and the mist layer was observed as a cloud layer.

The synoptic conditions between 1800 LST 5th August and 1200 LST 6th August are shown in Figure A2. An anticyclone
was centred to the northeast of New Zealand with a ridge extending to the southwest. The location of the ridge over the South

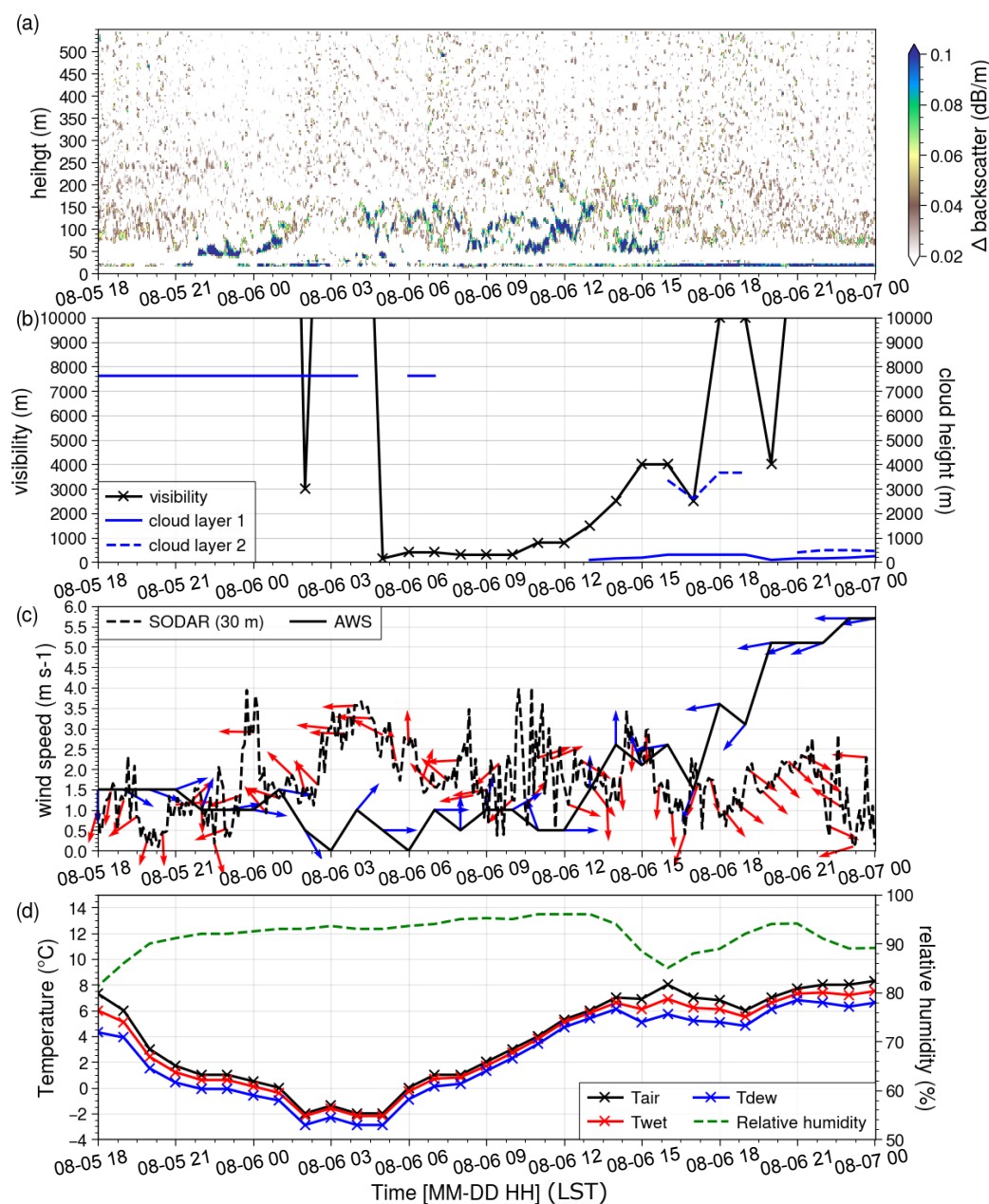

**Figure A1.** (a) Vertical time section of sodar backscatter, (b) time series of observed visibility and cloud height, (c) wind speed and direction taken from the AWS (10 m) at CHA and from a height of 30 m from sodar observations, (d) air temperature ($T_{air}$), wet bulb temperature ($T_{wet}$), dew point temperature ($T_{dew}$), and relative humidity from the AWS at CHA.

Island indicates less cloudy conditions. No significant precipitation was recorded within 5 days before the fog event (not shown)

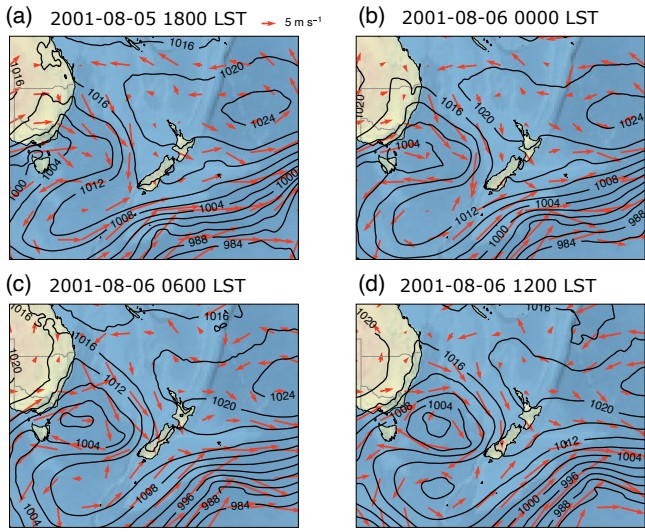

**Figure A2.** Mean sea level pressure (hPa) and 10 m wind vectors at (a) 1800 LST on 5th August 2001, and (b) 0000 LST, (c) 0600 LST, and at (d) 1200 LST on 6th August 2001. Data were obtained from ERA5 (Hersbach et al., 2019).

so that the state of soil moisture was not particularly high. Based on the evidence and the available data, this fog event was considered suitable to conduct radiation fog simulations to investigate our research questions.

## 575 Appendix B: Supporting information for soil moisture and analysis

### B1 Soil moisture heterogeneity adjustment method

An example of the readjustment mentioned in Section 5 is shown in Figure B1. For other adjustment of soil moisture values, the following method was applied to make all heterogeneous simulations have the same mean value as HOM. The mean soil moisture of HOM is denoted as $HOM_{mean}$ and the mean soil moisture of HET as $HET_{mean}$. The difference between the
580 mean values is $\Delta_{mean} = HET_{mean} - HOM_{mean}$. Then $\Delta_{mean}$ was added to each grid point to adjust the soil moisture value.

The adjustment is the same for the three-dimensional soil moisture field (west-east, south-north, and eight vertical levels of soil). For each soil layer, the difference between the domain mean value ($HOM_{mean}$) and the original mean soil moisture value ($SM_{mean}$) was first calculated as $\Delta_i = SM_{mean} - HOM_{mean}$ for each grid. Then $\Delta_i$ was added to each grid point at each soil layer.

### 585 B2 Accumulated latent heat flux

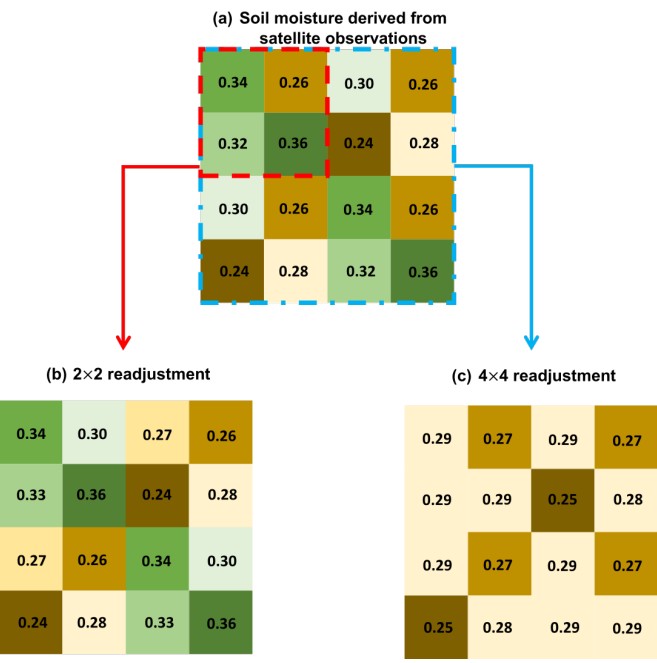

**Figure B1.** Examples of soil moisture readjustment: (a) demonstrates the soil moisture derived from the satellite observations. The mean value for (a) is 0.295 m3 m-3. Each square box represents one grid point, and the values of each box indicate the soil moisture content in m3 m-3. Boxes in green indicate that the soil moisture of the box is higher than the homogeneous soil moisture value, while boxes in brown indicate lower soil moisture values. A 2 by 2 area is highlighted by the red dashed square box and 4 by 4 area is highlighted by the blue dashed dotted square box in (a). (b) shows the results of 2 by 2 readjustment. (c) shows the results of 4 by 4 readjustment.

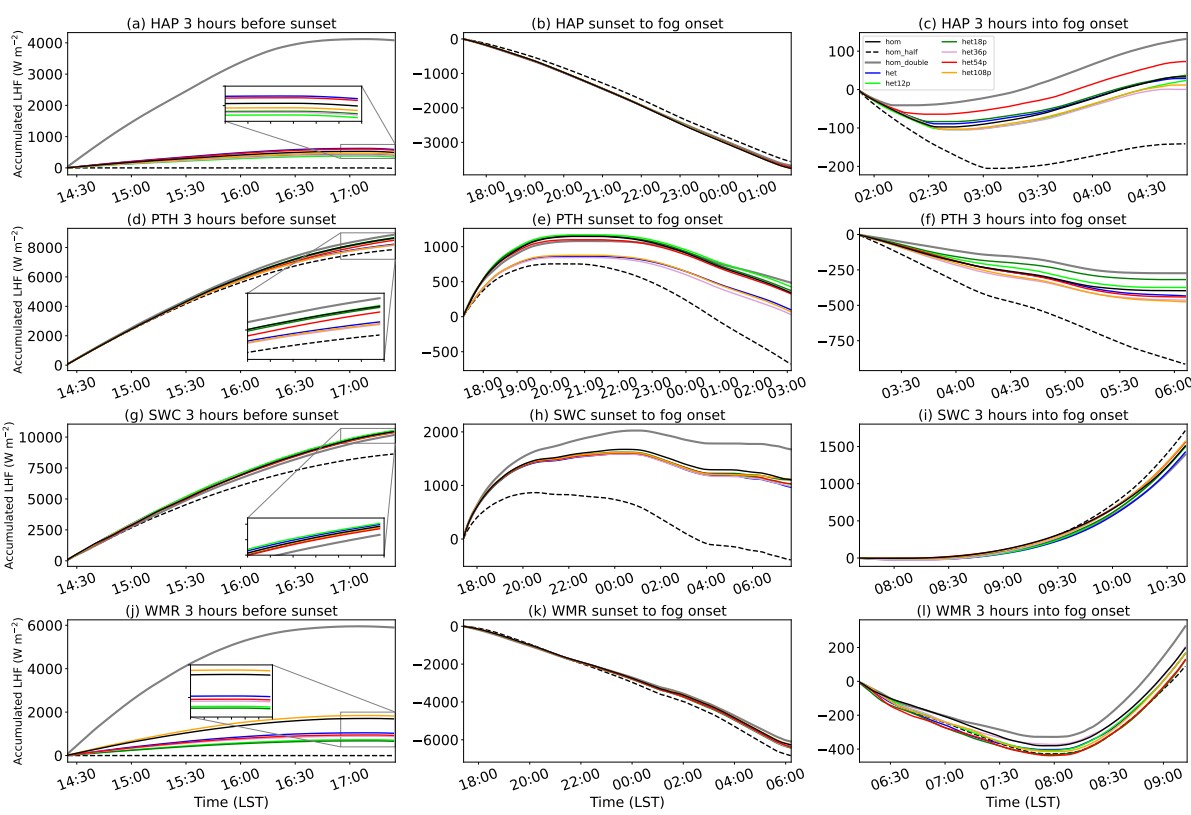

**Figure B2.** Accumulated latent heat flux (LHF) for the 3-hour period before sunset (a, d, g, j), the period between sunset and fog onset (b, e, h, k), and the 3-hour period into fog onset (c, f, i, l) for HAP, PTH, SWC, and WMR. Colours for all panels refer to panel (c) legend.

*Author contributions.* AS provided sodar data for analysis. DL was responsible for the data acquisition, setting up and running WRF and PALM simulations, and analysed the results. MK, LER, and BK supervised DL in designing and performing the case studies. DL wrote the manuscript with contributions from MK, LER, BK, and AS. DL, MK, LER, BK, and AS reviewed the manuscript.

*Competing interests.* The authors declare that they have no conflict of interest.

*Acknowledgements.* We would like to acknowledge New Zealand eScience Infrastructure (NeSI) high-performance computing facilities for providing computation resources for performing PALM simulations. We would like to thank Françios Bissey and the University of Canterbury high-performance computing cluster for providing computation resources for WRF simulations. The contribution of Dongqi Lin and Laura E. Revell was supported by the University of Canterbury and the Ministry of Business, Innovation and Employment project Particulate Matter Emissions Maps for Cities (grant no. BSCIF1802). Basit Khan received support from the MOSAIK and MOSAIK-2
projects, which are funded by the German Federal Ministry of Education and Research (BMBF) (grant nos. 01LP1601A and 01LP1911H), within the framework of Research for Sustainable Development (FONA; http://www.fona.de, last access: 26 October 2022). Marwan Katurji received support from the Royal Society of New Zealand (contract no. RDF-UOC1701).

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
