# Peer review of "Investigating multiscale meteorological controls and impact of soil moisture heterogeneity on radiation fog in complex terrain using semi-idealised simulations"

_EGUsphere, 2022_

## Referee Comment (RC1)

**Review of «Multiscale meteorological controls and impact of soil moisture heterogeneity on radiation fog in complex terrain" by Lin et al., submitted to EGUsphere**

By: Stephanie Westerhuis, January 2023

The manuscript by Lin at al. presents a modelling study on radiation fog around Christchurch, New Zealand. The authors design a fog case study combining WRF&PALM, remote-sensing observations of atmospheric profiles as well as satellite-derived soil moisture humidity. The study aims at answering two research questions with this experimental setup. 1) Which are the meteorological drivers for radiation fog in the Christchurch area? 2) What is the impact of soil moisture heterogeneity?

The first question is addressed by analysing near-surface water vapour and wind, fog duration and Modified Richardson number (an indicator for stability). The authors describe the spatial evolution of the nocturnal fog event. Fog formed mostly in regions where the regional wind systems did not prevent the formation of a near-surface stable layer.

The experiments targeting the second question comprise a set of simulations for which the soil moisture distribution has been adjusted to represent heterogeneity at different scales. While fog duration varies locally compared to the reference simulation, the spatial pattern does not change drastically and there is no obvious connection between the changes in soil moisture and the changes in fog duration. The authors conclude that the fog event is mostly dominated by larger-scale atmospheric drivers and regional conditions, but that the fog duration itself is determined by the micro-scale heterogeneities.

The authors have carefully designed a testbed for fog in complex terrain which holds the potential for valuable insights on fog processes. However, the analysis of the governing physical processes is incomplete and the conclusion that soil moisture heterogeneity dictates the fog duration lacks convincing arguments. The reader is left with the feeling that the authors had started off with a promising tool at hand and then stopped halfway to the finish line.

**Major items:**

- The authors make the strong statement that soil moisture heterogeneity is highly important for fog simulations but are not able to provide any explanation connecting the changes in the soil moisture to fog occurrence. They do not provide any convincing arguments that this is not just "noise". First of all, additional, **more detailed analyses of the driving processes could shed light on causalities** (more details below). Secondly, the **significance level of the changes in fog duration could be estimated** with more data (e.g. running the simulation for multiple days or creating an ensemble with slightly varying initial conditions).

- To provide more context for experiments focusing on complicated features such as heterogeneity **I recommend conducting some simpler sensitivity tests first**. It would be helpful to see the impact of doubling/halving the soil moisture in HOM. In a second step, the differences between HOM and HET should be addressed. And only after all the simpler aspects have been dealt with, the analysis should focus on the impact of heterogeneity itself.

- A **more detailed analysis of the modelled processes is required** for a complete understanding of the fog event.
    - How much soil moisture does actually end up in the atmosphere in different regions and for different model configurations? (Keyword: Accumulated latent heat flux.)
    - Figure 9 shows a significant low stratus cloud. It is only mentioned as "clouds that formed over this area" on L289. Low stratus clouds are often closely related to fog with respect to the driving processes (see the paper by Dupont, 2012, about the lifting of fog into low stratus), but a discussion of it is completely missing. There might be an interesting connection between the topography and/or advected air masses and the low stratus which forms in place of fog.
    - On L302, vertical entrainment is mentioned. In case PALM offers the option to output budget terms, it would be illustrative to see a budget of the resolved and parameterized turbulent processes at the cloud top.

- There is an **imbalance** between aspects mentioned in the abstract/introduction/methods (heterogeneous soil moisture, land use, topography) and those essentially presented in the Results. These comprise only one figure (Fig. 10) dedicated to the soil moisture heterogeneity, five figures (Figs. 5-9) elucidating the fog event evolution of the reference simulation, but e.g. the impact of land use is not discussed at all despite being mentioned multiple times in the previous sections. The text is lengthy but at the same time misses a lot of explanations. A revised version of the manuscript should **only include what is relevant for the specified research questions**.
    - E.g. there is no need to highlight impervious areas in Figure 1d when the study is not touching upon this subject.
    - E.g. L131: The sentence stating that "ground-based observations of visibility are necessary to conduct fog simulations" is first of all wrong. Observations are at most "useful" for the validation of such simulations. However, fog simulations could also be compared to surface observations of relative humidity and incoming LW radiation or with remote sensing devices which are frequently employed at airports such as ceilometers, and satellite imagery. Anyways, information about the feasibility of a comparison to observations is obsolete when the simulations are not connected to a specific real case anymore. Be clearer: Either the simulations are idealized, hence eliminate all of those passages, or argue why comparison with observations is justified and then also present this.
    - Another example is the explanation about two-moment microphysics which are not used after all (L213-215).

o   Furthermore, the Introduction and Section on Data Description evoke expectations to learn about the impact of heterogeneous land use which is not at all addressed in the Results. Conduct and present additional experiments or eliminate those parts.

o   Figure 6 includes a shading indicating the land use, while this is not discussed but instead hampers the interpretation of the displayed visibility.

- The **experimental setup is partially unclear**:
  o   How does the domain averaged soil moisture compare for the different heterogeneous setups to the homogeneous setup?
  o   How are the soil moisture profiles (i.e. the soil moisture below the first soil layer) adapted compared to HOM? → L193/4: Please explain what a "3D profile of soil moisture" is and how this is used. Please also explain how "an adjustment corresponding to the Landsat 8 observations" is conducted in detail.
  o   What about clouds in general at initialization and throughout the simulation?

- The **storytelling could be improved** if Methods and Results were not separated strictly. I would prefer reading about the setup of the reference simulation first, followed by the presentation of the event. Next, the changes made to the soil moisture were to be introduced and the results thereto follow afterwards.

- Figures should be refined: Some of them contain **features which are too small to interpret** (e.g. wind arrows in Fig. 9), many comprise axes labelling which is very small and would be easier to read if the employed units were km instead of m. Furthermore, some of the figure captions could be improved by adding more relevant information (e.g. which configuration does it refer to, e.g. Fig. 5) and removing irrelevant information (mentioning of impervious areas in Fig. 1).

- Personally, I have never made good experiences with calculating the horizontal visibility based on the liquid water content with the mentioned formula (1). Spatial plots of visibility often reproduce the distribution of the liquid water content. I see the benefit of being able to assign a "fog duration" to grid cells but for analyses of physical processes I have always found it **more revealing to analyse the prognostic model variables (liquid water content) directly**. Additionally, it is unclear whether the visibility is only calculated for the lowest model level or for a couple of levels close to the surface (L215).

**Detailed comments:**

As I expect the text to be revised substantially I have refrained from listing syntax and typing errors. The comments below aim to add onto the previously mentioned major items.

- Title: Specify that the study is about "modelling" and preferably also include the information that the experiments are "semi-idealised".
- Abstract, L1: The terms "fog development/persistence/formation" seem to be used imprecisely throughout the document. "Fog development" is somewhat vague, existing literature often

employs "fog formation" for the first stage of the fog life cycle or refers to the "fog life cycle" for all stages from formation until dissipation. E.g. L37: Do you mean the formation phase or the development after the fog has formed? L27: Is the intention indeed to highlight the importance of land surface characteristics on the "fog formation" instead of overall "fog occurrence"? I suggest to adopt the wording as used in the Introduction of the publication of Bergot and Lestringant (2019).

- Abstract, L6: Do not include links in the abstract.
- Abstract, L15: Readability would be improved if the units were chosen differently. I prefer reading "10 - 200km" instead of "10^4 – 2x10^5m. This holds true for the whole document, including the figures, which always use meters while axes in kilometers would be easier to interpret.
- L 26: Please do not cite a 30-year old publication to state that "land surface dynamics are the MOST IMPORTANT" factor. I would prefer to read something along the lines of "Already Duynkerke (1991) had identified land surface characteristics as one of the driving factors for fog".
- L 47: Suggestion to rephrase to "in a region in north-eastern France".
- L53: Could you add a reference to a publication addressing "changes in fog duration"?
- L64: Eliminate repetitions of non-domain-specific terms in consecutive sentences: "Aforementioned studies" appears again on L67. Please also keep an eye on this elsewhere in the text when revising.
- L68: "…did not consider effects of heterogeneity in land use or soil moisture" could be misinterpreted as "those studies are based on experiments with homogeneous land use and soil moisture". Could you please clarify that those studies did just not conduct sensitivity experiments specifically targeting this question?
- L69: Depending on how the additional work to provide a more in-depth analysis is addressed, this sentences could be re-phrased to "Therefore, the COMBINED effect of complex orography and heterogeneous soil moisture on…".
- L95: I suggest to prominently state that this is a "semi-idealised" numerical simulation and which makes use of "nesting".
- L175: Improve wording of "neither too high nor too low".
- L205: How exactly does the configuration of the outer domain "maintain a stable boundary layer"?
- L225: Specify: What is the soil moisture INDEX used for and how does it relate to the soil moisture values in kg/kg which are finally used in the experiments.
- It is unclear what is meant exactly with "fog onset". Does this time vary for each grid cell? If yes, I would be interested to see a spatial plot showing the timing of fog onset.
- L264: "greater qv" is not a proper expression. Greater than what exactly?
- L270: A "horizontal cross-section of the first model level" is just "the spatial distribution of xyz on the first model level".
- L271: The fog life cycle is strongly dependent on the incoming solar radiation (or lack thereof). I would prefer to have all mentioned time stamps in local time and a clear indication of sunrise and sunset times in the text as well as in the figures comprising a time axis.
- L276: Instead of indicating the location of the cross-section in meters I would prefer to see a vertical line drawn on e.g. Figure 5.

- L297: The storytelling of the reference simulation could be improved by focusing on different areas one after another.
- L307: Figure 9 is not a good reference to illustrate the surface elevation.
- L308: What is the causality between the "convergence of the southeasterly and northerly flow" and the "development of another stable layer"?
- L315: Does "in the meantime" refer to the period after sunrise?
- L323: On Figure 10 it is not evident whether there are large differences regarding the spatial distribution of fog OCCURRENCE between HOM and the HET experiments. It is left to the reader to determine the fog duration for a specific area from 10a and then subtract the delta in fog duration indicated in the other subfigures. An additional figure showing 3 categories (fog in both setups, fog only in HOM, fog only in HETxx) would be useful.
- L385: Specify: What would you expect to learn from a simulation at higher resolution with respect to soil moisture heterogeneity?

**References**

Bergot, T., & Lestringant, R. (2019). On the predictability of radiation fog formation in a mesoscale model: A case study in heterogeneous terrain. *Atmosphere*, *10*(4), 165.

Dupont, J. C., Haeffelin, M., Protat, A., Bouniol, D., Boyouk, N., & Morille, Y. (2012). Stratus–fog formation and dissipation: a 6-day case study. *Boundary-layer meteorology*, *143*(1), 207-225.

Duynkerke, P. G. (1991). Radiation fog: A comparison of model simulation with detailed observations. *Monthly Weather Review*, *119*(2), 324-341.

---

## Author Response (AR1)

**Authors' response for egusphere-2022-1229:**
**Multiscale meteorological controls and impact of soil moisture heterogeneity on radiation fog in complex terrain**

The authors thank the reviewers for their time and consideration given to this manuscript. The reviewer's comments have been listed below in *black italics* and responded to individually in *blue italics*. Revised sentences are in *red italics*.

We understand both the reviewers' concerns that more detailed analysis is necessary to better support our argument presented in this manuscript. The goal of this paper is to investigate the impact of soil moisture heterogeneity on fog. The main argument is that in this case study fog is highly controlled by mesoscale meteorology and topographical forcings, while the spatial heterogeneity in fog duration at microscale can be altered by soil moisture heterogeneity. We agree that the previous version of the manuscript may have failed to sufficiently and clearly deliver the scientific message. We have followed Dr Westerhuis' suggestions and have conducted two additional simulations of doubling/halving the soil moisture of the homogeneous setup. Analysis of fog spatial heterogeneity and pseudo-process diagrams have been added to the revised version of the manuscript. We have conducted a significant amount of analysis before the manuscript re-submission after receiving reviewers' comments, but no quantitative conclusions can be made due to the high heterogeneity and complexity of the simulations. Although idealised simulations mentioned by the reviewers can single out the processes involved, when all effects of surface heterogeneity were combined, the interactions between each process make definitive analysis difficult. However, more discussion has been added to the manuscript and the manuscript structure has been revised.

Due to technical issues, the data output file for the HET6p simulation was corrupted and cannot be used for analysis. In particular, after recent major maintenance of New Zealand's high-performance computing (HPC) system, we are experiencing technical issues running PALM and hence cannot currently re-run any of the fog simulations. We have therefore had to remove HET6p from the revised manuscript, although the exclusion of HET6p does not alter our conclusions.

**Reply to Stephanie Westerhuis:**

**Major items:**
- *The authors make the strong statement that soil moisture heterogeneity is highly important for fog simulations but are not able to provide any explanation connecting the changes in the soil moisture to fog occurrence. They do not provide any convincing arguments that this is not just "noise". First of all, additional, **more detailed analyses of the driving processes could shed light on causalities** (more details below). Secondly, the **significance level of the changes in fog duration could be estimated** with more data (e.g. running the simulation for multiple days or creating an ensemble with slightly varying initial conditions).*

*We agree with Dr Westerhuis that more detailed analysis could better support our argument presented in this manuscript. We therefore followed the reviewer's suggestion and conducted two additional simulations of doubling/halving the soil moisture in HOM. The two simulations are presented in the revised manuscript in Section 5.2. We have added more analysis using pseudo-process diagrams in respect to the driving processes in Section 5.2.3. Running simulations for multiple days or running an ensemble is not feasible due to the high computation cost. Each one of our current simulations already takes about 1 day of wall clock time on 244 Intel Xeon Skylake Gold 6148 processor cores, running at 2.4 GHz.*

- *To provide more context for experiments focusing on complicated features such as heterogeneity **I recommend conducting some simpler sensitivity tests first.** It would be helpful to see the impact of doubling/halving the soil moisture in HOM. In a second step, the differences between HOM and HET should be addressed. And only after all the simpler aspects have been dealt with, the analysis should focus on the impact of heterogeneity itself.*

*Before conducting the simulations presented in the papers, we did conduct a few sensitivity tests with/without flat terrain and with/without land use. While simple sensitivity tests allow us to interpret the processes easily, the analysis gets complicated when more heterogeneity is involved. The interactions between different processes make fog formation complex and this is why fog research is particularly difficult. We have added two additional simulations of doubling/halving the soil moisture in HOM (see Section 5.2), along with additional analysis and discussion (Sections 5.2.2, 5.2.3, and 5.2.4). The results support our argument in the original manuscript that soil moisture and its heterogeneity does not alter the general structure of fog but can impact fog duration.*

- *A **more detailed analysis of the modelled processes is required** for a complete understanding of the fog event.*
    - *How much soil moisture does actually end up in the atmosphere in different regions and for different model configurations? (Keyword: Accumulated latent heat flux.)*
    - *Figure 9 shows a significant low stratus cloud. It is only mentioned as "clouds that formed over this area" on L289. Low stratus clouds are often closely related to fog with respect to the driving processes (see the paper by Dupont, 2012, about the lifting of fog into low stratus), but a discussion of it is completely missing. There might be an interesting connection between the topography and/or advected air masses and the low stratus which forms in place of fog.*
    - *On L302, vertical entrainment is mentioned. In case PALM offers the option to output budget terms, it would be illustrative to see a budget of the resolved and parameterized turbulent processes at the cloud top.*

*We agree with the reviewer that more detailed analysis is required. To aid the understanding of the fog event, we have added more analysis regarding spatial heterogeneity of fog (Section 5 in the revised manuscript). We have conducted analysis on accumulated latent heat flux, but the results do not show a significant difference between HOM and heterogeneous simulations (Section 5.2.3 and Appendix B Figure B2). Changes in soil moisture and its heterogeneity do lead to changes in accumulated latent heat flux (Figure B2), but do not lead to an increase/decrease in liquid water mixing ratio (Figure 12). Screenshots of Figure B2 and Figure 12 are attached below. For better image quality, please refer to the manuscript pdf.*

[Figure]

**Figure B2.** Accumulated latent heat flux (LHF) for the 3-hour period before sunset (a, d, g, j), the period between sunset and fog onset (b, e, h, k), and the 3-hour period into fog onset (c, f, i, l) for HAP, PTH, SWC, and WMR. Colours for all panels refer to panel (c) legend.

[Figure]

**Figure 12.** Scatter plots showing Accumulated latent heat flux (LHF) against accumulated $ql$ coloured by soil moisture for HAP (a), PTH (b), SWC (c), and WMR (d). Data samples were obtained from each grid point at 1 km scale of all simulations. The accumulation period is 3 hours into fog onset.

*To accurately quantify the amount of soil moisture ending up in the atmosphere and advected to where fog occurred, we believe more analysis is required using a Lagrangian approach. However, the amount of work involved in Lagrangian analysis would require an additional research paper.*

*We agree that the clouds could be an interesting factor. However, in this manuscript, we only want to focus on the near-surface processes. Analysis of the role of low cloud could be part of future work.*

- *There is an **imbalance** between aspects mentioned in the abstract / introduction / methods (heterogeneous soil moisture, land use, topography) and those essentially presented in the Results. These comprise only one figure (Fig. 10) dedicated to the soil moisture heterogeneity, five figures (Figs. 5-9) elucidating the fog event evolution of the reference simulation, but e.g. the impact of land use is not discussed at all despite being mentioned multiple times in the previous sections. The text is lengthy but at the same time misses a lot of explanations. A revised version of the manuscript should **only include what is relevant for the specified research questions**.*
  - *E.g. there is no need to highlight impervious areas in Figure 1d when the study is not touching upon this subject.*

*We have revised the Figure 1 caption as follows:*
*Maps of the case study simulation domains: (a) a New Zealand topographic map with a red square indicating the location of the simulation domains, (b) a topographic map (height above sea level) showing the simulation domain configuration for the case study, (c) a topographic map of simulation domains 3 and 4 (D03 and D04), the locations of the FENZ (Fire and Emergency New Zealand) weather stations, and the location of the sodar operated at Christchurch airport, and (d) a land use map of D04 with buildings and streets in white. The simulation domain 1 (D01) configured as flat terrain with short grass, and hence no topography, is shown in panel (b). The logarithmic topographic height colormap in panel (b) applies to panels (b) and (c), while grey indicates the ocean.*

*We consider the illustration of heterogeneous land use important because such heterogeneity has led to spatial heterogeneity in fog. Additional discussion regarding land use has therefore been added in Section 5.2.3 (between Line 408 ad Line 412) in the revised manuscript.*

  - *E.g. L131: The sentence stating that "ground-based observations of visibility are necessary to conduct fog simulations" is first of all wrong. Observations are at most*

*"useful" for the validation of such simulations. However, fog simulations could also be compared to surface observations of relative humidity and incoming LW radiation or with remote sensing devices which are frequently employed at airports such as ceilometers, and satellite imagery. Anyways, information about the feasibility of a comparison to observations is obsolete when the simulations are not connected to a specific real case anymore. Be clearer: Either the simulations are idealized, hence eliminate all of those passages, or argue why comparison with observations is justified and then also present this.*

*We agree with the reviewer that the information regarding observations and field campaigns is redundant for this case study. We have revised the manuscript, moved the discussion of observations to Appendix A, and acknowledged that this study is "semi-idealised". The simulations are not completely idealised as the initialisation profiles of atmosphere were obtained from observational data. Realistic geospatial data were used in the simulations. However, the simulations are not realistic that they were not conducted for the purpose of replicating a real fog event. In addition, the simulation results were not validated using observational data.*

- o *Another example is the explanation about two-moment microphysics which are not used after all (L213-215).*

*The explanation about two-moment microphysics has been removed in the revised manuscript.*

- o *Furthermore, the Introduction and Section on Data Description evoke expectations to learn about the impact of heterogeneous land use which is not at all addressed in the Results. Conduct and present additional experiments or eliminate those parts.*

*Inclusion of contents related to land use in the Introduction and Data Description is to show that heterogeneous land use is considered in this study. Additional discussion regarding land use has been added in Section 5.2.3 (between Line 408 ad Line 412) in the revised manuscript.*

- o *Figure 6 includes a shading indicating the land use, while this is not discussed but instead hampers the interpretation of the displayed visibility.*

*We have revised Figure 6 (now Figure 4 in the revised manuscript, see figure attached below). The shading of land use has been removed. The units of axes are now in km. Liquid water mixing ratio was plotted instead of visibility. Topography height contours were added. Density of wind vectors has been decreased for better visualisation. Timestamps are in local standard time (LST).*

[Figure]

- *The **experimental setup is partially unclear**:*

  - *How does the domain averaged soil moisture compare for the different heterogeneous setups to the homogeneous setup?*

*The domain averaged soil moisture is identical in the homogeneous setup and all the heterogeneous setups. This has been clarified in the revised manuscript (Section 5.2.1):*
*The spatially heterogeneously distributed soil moisture of the simulation domain was first adjusted to have the same mean value as HOM. The soil moisture was then adjusted to amplify the signal of soil moisture heterogeneity. Finally, the soil moisture at each grid point was adjusted again such that the mean value of the entire domain is identical to HOM.*

  - *How are the soil moisture profiles (i.e. the soil moisture below the first soil layer) adapted compared to HOM? → _L193/4: Please explain what a "3D profile of soil moisture" is and how this is used. Please also explain how "an adjustment corresponding to the Landsat 8 observations" is conducted in detail.*

*To clarify the adjustment method used for soil moisture, we have added the following details in Appendix B1 of the revised manuscript:*

*An example of the readjustment introduced in Section 5 is shown in Figure B1. For other adjustment of soil moisture values, the following method was applied to make all heterogeneous simulation shave the same mean value as HOM. The mean soil moisture of HOM is denoted as HOMmean and the mean soil moisture of HET as HETmean. The difference between the mean values is Δmean = HETmean −HOMmean. Then Δmean was added to each grid point to adjust the soil moisture value.*

*The adjustment is the same for the three-dimensional soil moisture field (west-east, south-north, and eight vertical levels of soil). For each soil layer, the difference between the domain mean value (HOMmean) and the original mean soil moisture value (SMmean) was first calculated as Δi = SMmean −HOMmean for each grid. Then Δi was added to each grid point at each soil layer.*

  o *What about clouds in general at initialization and throughout the simulation?*
*All simulations presented in the manuscript have clear sky at initialisation and throughout the period before fog formation. Clouds only appeared around 0100 LST (Figure 6 in the revised manuscript, attached below). The cloud layer lifted after sunrise, but patchy clouds were still present in the simulation until late afternoon. This has been clarified in the manuscript as follows:*
*The simulation had clear sky from initialisation (not shown). The cloud and fog started to lift and dissipate around sunrise (0736 LST), while patchy clouds were still present 3 hours after sunrise.*

[Figure]

- *The **storytelling could be improved** if Methods and Results were not separated strictly. I would prefer reading about the setup of the reference simulation first, followed by the presentation of the event. Next, the changes made to the soil moisture were to be introduced and the results thereto follow afterwards.*
*We have followed the reviewer's suggestion and have revised the manuscript as follows:*
  1. *Introduction*
  2. *Data description*
  3. *Model and simulation configuration*
  4. *Meteorological controls in the baseline scenario*
  5. *Investigation of the impact of soil moisture heterogeneity*
     *5.1 Soil moisture heterogeneity configuration*
     *5.2 Results and analysis*
     *5.3 Discussion*
  6. *Conclusions*
*For more details, please refer to the revised manuscript.*

- *Figures should be refined: Some of them contain **features which are too small to interpret** (e.g. wind arrows in Fig. 9), many comprise axes labelling which is very small and would be easier to read if the employed units were km instead of m. Furthermore, some of the figure captions could be improved by adding more relevant information (e.g. which configuration*

*does it refer to, e.g. Fig. 5) and removing irrelevant information (mentioning of impervious areas in Fig. 1).*

*All figures have been refined following this suggestion. All units have been changed from m to km. Plots with wind arrows have been refined for better visualisation and readability. All figure captions have been revised.*

- *Personally, I have never made good experiences with calculating the horizontal visibility based on the liquid water content with the mentioned formula (1). Spatial plots of visibility often reproduce the distribution of the liquid water content. I see the benefit of being able to assign a "fog duration" to grid cells but for analyses of physical processes I have always found it **more revealing to analyse the prognostic model variables (liquid water content) directly**. Additionally, it is unclear whether the visibility is only calculated for the lowest model level or for a couple of levels close to the surface (L215).*

*We agree with the reviewer that analysis of liquid water mixing ratio (ql) directly is more revealing. Visibility calculated using Equation 1 (vis =0.02 \* LWC^-0.88, in the original manuscript) usually reaches < 1 km once ql > 0. We therefore decided to use liquid water mixing ratio directly in the revised manuscript. The analysis of this manuscript only focuses on the lowest model level. We have clarified this as follows (Line 200 in the revised manuscript):*

*Fog is identified when liquid water is present at the first model level, i.e. when liquid water mixing ratio (ql) is greater than zero.*

**Detailed comments:**

*As I expect the text to be revised substantially I have refrained from listing syntax and typing errors. The comments below aim to add onto the previously mentioned major items.*

- *Title: Specify that the study is about "modelling" and preferably also include the in that the experiments are "semi-idealised".*

*The title of this paper has been revised as follows:*

*Investigating multiscale meteorological controls and impact of soil moisture heterogeneity on radiation fog in complex terrain using semi-idealised simulations*

- *Abstract, L1: The terms "fog development/persistence/formation" seem to be used imprecisely throughout the document. "Fog development" is somewhat vague, existing literature often employs "fog formation" for the first stage of the fog life cycle or refers to the "fog life cycle" for all stages from formation until dissipation. E.g. L37: Do you mean the formation phase or the development after the fog has formed? L27: Is the intention indeed to highlight the importance of land surface characteristics on the "fog formation" instead of overall "fog occurrence"? I suggest to adopt the wording as used in the Introduction of the publication of Bergot and Lestringant (2019).*

*We have revised the manuscript and adopted the wording based on Bergot and Lestringant (2019).*

*Abstract Line 1:*

*Coupled surface-atmosphere high-resolution simulations were carried out to understand meteorological processes involved in radiation fog life cycle in a city surrounded by complex terrain.*

*Line 27 (now Line 28 in the revised manuscript):*

*Over 30 years ago, Duynkerke (1991) identified land surface physical characteristics as the most important factor for fog occurrence, among several thermal and dynamical processes in the ABL*

*Line 37:*

*Over the past decade, many studies have included heterogeneous land surface characteristics in radiation fog simulations in order to understand the microscale processes (occurring from 1 cm to 1 km, and from seconds to hours) and associated feedback during fog life cycle.*

- *Abstract, L6: Do not include links in the abstract.*

*The link has been removed.*

- *Abstract, L15: Readability would be improved if the units were chosen differently. I prefer reading "10 - 200km" instead of "10^4 – 2x10^5m. This holds true for the whole document, including the figures, which always use meters while axes in kilometers would be easier to interpret.*

*We have revised the units in the manuscript and changed axes in metres to kilometres. For example, Line 16-17, Line 36, and Line 227.*

- *L 26: Please do not cite a 30-year old publication to state that "land surface dynamics are the MOST IMPORTANT" factor. I would prefer to read something along the lines of "Already Duynkerke (1991) had identified land surface characteristics as one of the driving factors for fog".*

*This sentence has been rephrased as follows:*
*Over 30 years ago, Duynkerke (1991) identified land surface physical characteristics as the most important factor for fog occurrence, among several thermal and dynamical processes in the ABL*

- *L 47: Suggestion to rephrase to "in a region in north-eastern France".*

*This has been rephrased as follows:*
*… in a region in north-eastern France*

- *L53: Could you add a reference to a publication addressing "changes in fog duration"?*

*Most papers do not address "fog duration" directly and they focus more on fog formation and dissipation. We hence have revised this sentence and added citations as follows:*
*Changes in soil moisture lead to variability in the surface energy balance, and consequently changes in fog formation and dissipation times (e.g. Bergot and Guedalia 1994, Remy and Bergot 2009, and Maronga and Bosveld 2017).*

- *L64: Eliminate repetitions of non-domain-specific terms in consecutive sentences: "Aforementioned studies" appears again on L67. Please also keep an eye on this elsewhere in the text when revising.*
- *L68: "…did not consider effects of heterogeneity in land use or soil moisture" could be misinterpreted as "those studies are based on experiments with homogeneous land use and soil moisture". Could you please clarify that those studies did just not conduct sensitivity experiments specifically targeting this question?*

- *L69: Depending on how the additional work to provide a more in-depth analysis is addressed, this sentences could be re-phrased to "Therefore, the COMBINED effect of complex orography and heterogeneous soil moisture on…".*

*We agree that these sentences are lengthy and can be misleading. We have revised Line 64-70 as follows:*
*However, to the best of our knowledge, no study has investigated the impact of soil moisture heterogeneity on radiation fog in combination with heterogeneous land use and topography.*

- *L95: I suggest to prominently state that this is a "semi-idealised" numerical simulation and which makes use of "nesting".*

*This has been revised as follows:*
*A radiation fog scenario was created in semi-idealised numerical simulations containing nested domains with the finest grid spacing of 81 m.*

- *L175: Improve wording of "neither too high nor too low".*

*We have reworded "neither too high nor too low" to suitable.*

- *L205: How exactly does the configuration of the outer domain "maintain a stable boundary layer"?*

*The flat parent domain was used as a buffer domain. When cyclic boundary conditions are used with steep terrain near the lateral boundary, there could be instability at the lateral boundaries where*

*inflow/outflow directions could be opposite. This may prevent a stable near-surface layer developing, while extra tests and investigation may be needed to verify that the configuration is responsible for maintaining a stable boundary layer. In the simulation, the evolution of the boundary layer has its own diurnal cycle. Therefore, we have revised this sentence as follows:*

*The purpose of this domain is to pass down the synoptic forcing to the finer domains and to avoid numerical instability caused by steep terrain near the periodic lateral boundaries.*

- *L225: Specify: What is the soil moisture INDEX used for and how does it relate to the soil moisture values in kg/kg which are finally used in the experiments.*

*As this study mainly focuses on soil moisture heterogeneity instead of accurate soil moisture content itself, we directly used the derived soil moisture index (SMI) as a measure of soil moisture for the simulations. The conversion between SMI and soil moisture requires a known wilting point and field capacity, for which extra work would be needed. We have added more information regarding SMI as follows:*

*To include soil moisture heterogeneity in the simulations, this study adopted the soil moisture index (SMI) calculation method for Landsat 8 imagery described in Avdan and Jovanovska (2016) and Potic et al. (2017). SMI describes the proportion of actual soil water content relative to a known wilting point and field capacity (e.g. Zeng et al. 2004, Hunt et al. 2009). Our case study focuses on the spatial heterogeneity rather than accurate soil moisture values. Therefore, in this paper the derived SMI is used as a surrogate for actual soil moisture.*

- *It is unclear what is meant exactly with "fog onset". Does this time vary for each grid cell? If yes, I would be interested to see a spatial plot showing the timing of fog onset.*

*In the original manuscript, "fog onset" is recognised at a grid point, when visibility at the first modelled level became less than 1 km. In the revised version, we define fog onset as when ql>0 was first detected at a grid point at the lowest model level. The fog onset time varies for each grid cell. Figures of the timing of fog onset are attached below:*

[Figure]

*Figure AC1. Fog onset time at each grid point for HOM. Timestamps are in LST.*

[Figure]

*Figure AC2. Fog onset time difference between HOM and all other simulations. Red indicates formation is delayed in HOM and blue indicates the opposite.*

*Due to the significant spatial heterogeneity of fog onset time, dissipation time, and duration, we decided to only focus on four specific sites (see the revised manuscript for more details) instead of*

*the entire simulation domain. We recognise fog formation time of a chosen area when fog coverage of the area rises from 5% to 95%, following the definition by Bergot and Lestringant (2019).*

- *L264: "greater qv" is not a proper expression. Greater than what exactly?*

*We have revised this as follows:*

relatively high qv (> 5.0 g kg$^{-1}$)

- *L270: A "horizontal cross-section of the first model level" is just "the spatial distribution of xyz on the first model level".*

*This has been revised as follows:*

To provide more insight into the fog simulation, the spatial distribution of ql and MRi on the first model level…

- *L271: The fog life cycle is strongly dependent on the incoming solar radiation (or lack thereof). I would prefer to have all mentioned time stamps in local time and a clear indication of sunrise and sunset times in the text as well as in the figures comprising a time axis.*

*We have revised the manuscript and all time stamps are now in local standard time (LST).*

- *L276: Instead of indicating the location of the cross-section in meters I would prefer to see a vertical line drawn on e.g. Figure 5.*

*A vertical line has been added on Figure 5 (now Figure 3 in the revised manuscript, attached below).*

[Figure]

- *L297: The storytelling of the reference simulation could be improved by focusing on different areas one after another.*

*We have revised the entire section (Section 4 of the revised manuscript) focusing on four selected areas - Hagley Park (HAP), Port Hills (PTH), Southwest of Christchurch (SWC), and Waimakariri River (WMR).*

- *L307: Figure 9 is not a good reference to illustrate the surface elevation.*

*We have revised Figure 9 (now Figure 6 in the revised manuscript, attached below) and a logarithmic scale is now used for the height axis to illustrate surface elevation.*

[Figure]

- *L308: What is the causality between the "convergence of the southeasterly and northerly flow" and the "development of another stable layer"?*

*This is similar to the drainage zone illustrated in the publication of Corsmeier et al. (2006), where air is stagnant and turbulence suppressed, and therefore stable layer can develop. This has been revised as follows:*

*Another drainage zone developed due to flow convergence, where turbulence was suppressed and a stable layer started to grow.*

- *L315: Does "in the meantime" refer to the period after sunrise?*

*Yes. To make this clear, we have replaced "In the meantime" with After sunrise*

- *L323: On Figure 10 it is not evident whether there are large differences regarding the spatial distribution of fog OCCURRENCE between HOM and the HET experiments. It is left to the reader to determine the fog duration for a specific area from 10a and then subtract the delta in fog duration indicated in the other subfigures. An additional figure showing 3 categories (fog in both setups, fog only in HOM, fog only in HETxx) would be useful.*

*We agree that it is difficult to determine differences in spatial distribution of fog occurrence. A figure (Figure 9 in the revised manuscript, attached below) has therefore been added in the revised manuscript to illustrate the spatial distribution of fog occurrence.*

[Figure]

Fog only in HOM                                        No fog in HOM

- *L385: Specify: What would you expect to learn from a simulation at higher resolution with respect to soil moisture heterogeneity?*

*A finer grid spacing may reveal more regarding how soil moisture interacts with the atmosphere under different land use. At current grid spacing (81 m), the land use is extrapolated. The plant and urban canopy are not well represented. A better resolved plant canopy may give more results regarding evapotranspiration so that identifying the contribution of soil moisture may be less difficult.*

*If observational data are available, it would be possible to use finer grid spacing to better resolve the small turbulent eddies and better simulate and forecast fog.*

**Reply to Anonymous Referee #2:**

**Major concerns:**

- *The numerical choices on the model configuration are not always clearly presented, and are sometimes debatable (see more details below). This does not give confidence in the quality of the results.*

*We thank the reviewer for the detailed comments. Our response to the concern regarding the technical specifications of the simulations is listed below.*

- *The first part of the results concerning the meteorological controls is confusing, explanations are difficult to understand in relation to the geography of the maps, and we do not understand the results that emerge. I really found the 4.1 part very difficult to read: it is not always clear which part of the map is commented on.*

*We have revised Section 4 so that the entire section could be more readable. All figures have been revised for better visualisation and readability. We have selected four areas to focus on and added labels for location references - Hagley Park (HAP), Port Hills (PTH), Southwest of Christchurch (SWC), and Waimakariri River (WMR). For details, please refer to the revised manuscript (Sections 4 and 5).*

- *The conclusion of this part is that the macrostructure of fog occurrence and distribution is highly controlled by topography and the mesoscale meteorology: although we are not very surprised by this conclusion, it is a bit hasty and we do not learn much.*

*We understand that the original version of the manuscript may lack good story telling, and we have revised the manuscript following helpful suggestions from both the reviewers.*

- *There is no comparison to observations, which would allow to know if the simulations are relevant and can be considered as reliable. We understand that authors do not attempt to replicate a real radiation fog event in the simulations. But a direct comparison of the simulation to the observations presented in A1 would have been beneficial to give confidence in the simulations, as well as a comparison of cloud fraction or cloud water content to satellite images.*

*We understand the value of using observations to verify the simulations. However, as stated in Appendix A, it is not possible for us to do such comparison with the very limited observational network of Christchurch. Therefore, we have clarified that this study is semi-idealised, and unfortunately no direct comparison between simulations and observations can be undertaken due to this limitation.*

- *For the second part of the results concerning different soil moisture conditions, the test only focuses on the magnitude of soil moisture heterogeneity. But a preliminary test would be necessary, dealing with the sensitivity to the soil moisture itself.*

*Fog sensitivity to soil moisture has been studied particularly using PALM by Maronga and Bosveld (2017). We have added two extra simulations with doubled and halved soil moisture compared to HOM to show the fog sensitivity to soil moisture. More details are provided in the revised manuscript, Section 5.2.*

- *Results concerning soil moisture heterogeneity do not show a clear impact on the fog duration. Indeed, the authors first underline that "there is no direct evidence to link the changes in soil moisture to the changes in fog duration" but just after they conclude that fog duration is sensitive to changes in soil moisture heterogeneity at microscale: here again, this is not really acceptable.*

*We understand that more analysis is required to support our argument. Following helpful suggestions provided by the reviewers, we have carried out more analysis and the results are presented in the revised manuscript.*

- *The presented fields are rather poor compared to the potential of diagnostics classically available in a LES (vertical temporal evolution, budgets ...). For instance, the fog life cycle is*

*only represented through fog duration 2D maps, without separating initiation and dissipation times.*

*We would like to clarify that our simulations are not typical LES. Rather they are high resolution mesoscale simulations (see Cuxart 2015). As the simulations include heterogeneous land use, topography and soil moisture, it is difficult to present vertical temporal evolution of fog as each grid cell has its own unique profile. In this study, we want to focus on the spatial distribution of fog near the surface. To address the reviewers' concerns, we have revised the manuscript and added more analysis on the temporal evolution of fog and the processes involved.*

- *In the same way, the paper deals with the impact of soil moisture, but no surface heat flux is presented.*

*We have added more analysis on accumulated latent heat flux and revised the manuscript accordingly to address the lack of data analysis.*

**Model configuration:**

*The choices of simulation configuration are not clearly enough presented and argued, with sometimes questionable choices:*

- *It seems that the first vertical level is at 18 m height: it is not suitable at all to fog simulations as we know the necessity to have a first level very close to the ground (max 2 m) (see Tardif, 2007)*

*First, we would like to point out that the first vertical level is at 9 m above the ground instead of 18 m. The vertical grid spacing is 18 m. We understand that it is essential to have fine grid spacing close to the ground for numerical fog forecasting. While ideally we would like to aim at finer grid spacing to better resolve all processes involved in fog, it is extremely expensive to conduct simulations for such a big domain. Each one of our current simulations already takes about 1 day of wall clock time on 244 Intel Xeon Skylake Gold 6148 processor cores, running at 2.4 GHz. Therefore, we have clarified in the manuscript that our simulations are high resolution mesoscale simulations rather than real large eddy simulations (LES).*

*We only aim to investigate the impact of spatial soil moisture heterogeneity within a complex environment using the high-resolution mesoscale approach and are hoping this could provide guidance for fog forecasting. We understand this is a limitation of this study which we have discussed in Section 6.*

- *The initialisation of PALM mixes observed atmospheric vertical profiles (for U, V, q) and simulated ones (for rv) from WRF: are they consistent? In other words, are the WRF simulated U, V, q profiles close to the observed ones?*

*We would like to clarify that this study does not aim to replicate a specific fog event, as it is a semi-idealised study. The vertical profiles of temperature, U, and V were obtained from observational data. The observational data do not include any vertical profile of q. The vertical profile of qv (water vapour mixing ratio) was obtained from WRF simulations because no observational data are available for Christchurch. It is not clear what the rv variable that the reviewer referred to is?*

*The initialisation profiles were carefully selected so that radiation fog can be simulated in PALM. We did carry out a comparison between WRF and the observations regarding the vertical profiles of winds and temperatures as shown in the figures below:*

[Figure]

In these figures, 'sodar' indicates observations obtained from the sodar deployed at the airport, AWS indicates data obtained from the automatic weather station located near the sodar, T2 is air temperature at 2 m simulated by WRF, and Tair indicates air temperature at various levels. The upper air observations include two sets of data: one for temperature and one for wind. The temperature data set includes vertical profiles of temperature only, while the wind data set includes vertical profiles of both wind and temperature. In general, WRF shows good agreement with all the observations.

- *I suppose that D01 is used without orography for the cyclic boundary conditions: this should be indicated.*

Yes, it is correct that D01 uses flat topography, which was stated in the manuscript in the Figure 1 caption, model configuration (Table 3), and Line 203 in the original manuscript (Line 188 in the revised manuscript).

- *what turbulence scheme is used? What are its characteristics (order of closure, mixing length, 3D or 1D …)?*

We used PALM's default 1.5 order scheme for turbulence closure, and for more details of this scheme refer to https://palm.muk.uni-hannover.de/trac/wiki/doc/tec/sgs and Maronga et al. (2015). We have added more description in Line 165 as follows:

All simulations used a modified three-dimensional Deardorff 1.5-order turbulence closure scheme, in which the energy transport by sub-grid scale eddies is assumed to be proportional to the local

*gradient of the average quantities (Deardorff 1980; Moeng and Wyngaard 1988; Saiki et al. 2000; Maronga et al. 2015).*

- *why is the microphysical scheme off in D02 and D03? Is it a way to prevent cloud advection from D02 to D03?*

*Yes, the microphysical scheme was switched off in D02 and D03. This is to simplify the processes involved in fog formation because the simulation in its current form is already very complex to analyse. If both D02 and D03 have the microphysical scheme turned on, it would be even more difficult to understand the impact of soil moisture heterogeneity. In this semi-idealised study, we want to understand the processes involved in D04 first before conducting more complex simulations.*

- *for the Kessler scheme, what constant value is used for droplet concentration? Is there droplet settling and if yes, how is it implemented? Is there droplet deposition?*

*The Kessler scheme embedded in PALM is a simplified one-moment bulk cloud scheme which does not include any microphysical cloud droplet concentration in the parameterisation. This scheme simply automatically converts supersaturation into liquid water without any prognostic quantities for microphysical variables. Cloud water sedimentation was enabled. For clarification, the following sentence has been added in the revised manuscript:*

*Cloud water sedimentation based on Ackerman et al. (2009) is enabled.*

*Ideally, one would aim at using a two-moment microphysical scheme, but this is not allowed in PALM when there is a plant canopy in the simulation, as described in Section 3.*

---

## Referee Report (RR1)

Manuscript number: egusphere-2022-1229
Title: Investigating multiscale meteorological controls and impacts of soil moisture heterogeneity on radiation fog in complex terrain using semi-idealised simulations
Authors: Dongqi Lin et al.

Reviewer recommendation: *rejection*

**Summary:**
The authors present a modeling study focusing on radiation fog in a complex terrain using a multi-nesting model domain employing the models WRF and PALM. The topic is highly relevant, since the different mechanisms interacting in fog are not well understood, in particular those related to surface heterogeneity. However, I found severe flaws in the methodology. In particular, the authors use a large-eddy simulation (LES) code at grid spacings (horizontally 729 – 81 m, vertically 32 – 18 m) that violate the constraints of the technique. The full turbulence spectrum, and the inertial subrange in particular, are unresolved in all simulations presented in the manuscript. All turbulent transport thus is sub-grid scale, which is not allowed in LES modeling as the filtering length must be within the inertial subrange and the bulk part of the turbulence kinetic energy must lie in the resolved scales. The authors did not make any effort to discuss this flaw. As a direct consequence the presented results are not reliable and I did not make the effort to read the discussion of the results in details. In the discussion section they report different results than found by a previous study (Maronga & Bosveld, 2017), both using the same PALM model system. It is likely that these differences are simply due to insufficient grid resolution. As was pointed out by Maronga & Bosveld, the required grid spacing for a typical radiation fog event was 1 m (both vertically and horizontally). The authors here have at best a factor 18 coarser grid spacing. There are more issues, which I detail below. Given this severe flaw, which cannot be removed unless the authors use extremely higher resolution and repeat all analyses, I recommend to reject the paper.

Detailed comments are given below.

**Detailed comments by the reviewer:**

Major comments

1.  A four-step multiple self-nesting of PALM is employed, with grid spacings between 729 m and 81 m horizontally and 162 m  to 18 m vertically. As outlined in the general summary: none of these grid spacings are sufficient to resolve the turbulence in a typical environment prone to radiation fog. Even for a convective boundary layer, where the dominant eddies are large, the coarsest grid spacing allowed is around 100 m. The grid spacings used here in the domains D01-D04 are way beyond what is possible to use in an LES model. Under stable conditions, the largest eddies are usually not larger than 10 m, so the grid spacing must be way smaller than that. You either need much higher grid spacings (in LES of radiation fog, grid spacings in literature are in the order of 1 - 4 m horizontally and vertically!), or you need to use a RANS model. By violating the constraints of LES, you are parameterizing all turbulent transport with a subgrid-scale model, which assumes to only treat small-scale isotropic turbulent fluxes. As this is not the case, the transport will be totally wrong. There is no discussion about this in the paper, except one sentence, saying that most of the turbulence is parameterized. Furthermore, no vertical profiles and turbulent quantities are presented. One might suspect this is because they will immediately show these flaws. If the authors cannot correct for these flaws, they probably better go for a RANS model where the grid spacing issue is somewhat less severe (however, to resolve fog layers, small vertical grid spacings are still essential!).

2.  You report you are using RRTMG as radiation code, but you also refer to have complex terrain and buildings in the domain. As RRTMG is operating as a single vertical column model, how do you calculate radiative fluxes at non-horizontal surfaces? As far as I know, PALM automatically uses a radiative transfer scheme (RTM) as soon as buildings or complex terrain is found in the domain. RTM,

however, cannot consider clouds and only works for clear-sky conditions. Also, it does not calculate flux divergences, which play a key role in fog development. I found no statement on how this problem is treated in the study.

3. Why is the most simply cloud microphysics available in PALM used? By default, PALM uses a two-moment scheme, which is kind of a standard for years. Is there any reasonable argument for switching to a simplistic Kessler scheme? Furthermore, cloud physics are only allowed in the D04 domain, which means that fog cannot be advected in the D04 domain. Does that make sense? Also, this means that there can be supersaturated air inside the D01-D03 domains. If this air is advected into D04 it will lead to spurious condensation.

4. The authors did a purely idealized study with no relation to any observed fog case. While I would agree that this might not be overly critical, in this particular case it makes me worry. As a combination of the technical flaw, the reader cannot evaluate whether the obtained results are by any means realistic.

---

## Referee Report (RR2)

The authors have made substantial efforts to improve the first version of the manuscript. The additional figures and analyses (e.g. pseudo-process diagrams) as well as the restructuring of the sections are well made. I am impressed by how much the text has improved and consider this work as valuable to the fog modelling community after a few minor revision points which are listed below have been addressed.

1. L18 and 38: As the model's horizontal resolution is on the order of several decametres, I would rather specify the microscale as "on the order of 100m to 1km".
2. L159: "This means THAT…"
3. L178: It is not self-explanatory what is meant by a "3D profile in west-east, south-north, and the vertical direction". I would replace the term "profile", as this in itself is associated with an extract of a quantity along the vertical dimension. Do you not just mean "soil moisture which varies both in the horizontal and vertical"?
4. L193 and 198: Repetition of "bulk cloud model only enabled in D04."
5. L199: "Cloud water sedimentation IS based on…"
6. L213: Improve wording of "fog event is the most significant".
7. L223: Swap ending of sentences: "..cross sections. The sunset time…on Day 1 and sunrise…"
8. L227: Either "temperature decreases" or "air cools"
9. L237: no "9" in "03900 LST".
10. L253: "High qv was PRESENT…"
11. L254 – 263: I find the explanation that clouds instead of fog formed very convincing and would replace "This is suspected to be due to" with "This is likely caused by". I would however specify the clouds to be "low stratus" which is often related to fog occurrence. The physical drivers could be more precise: "The layer of low stratus reflecting the outgoing LW radiation results in a reduced surface cooling and.." E.g. on the Swiss Plateau, the transition from fog to low stratus (and back) is often associated to increase (decrease) in wind speed. Is this also the case here?
12. L263: Suggestion: "Besides the low stratus clouds the simulation featured clear-sky conditions."
13. L331: T missing in "HET12p"
14. Figure 11: Specify not only the black solid line but also the grey shading in the first two columns.
15. L389: "Develop INto"
16. L408 and 409: The reasoning is not logically consistent:
    a) High LHF = high QL
    b) High soil moisture = high LHF
    c) High soil moisture =/ high QL
    -> Probably a) should be phrased differently.
17. L420: I suggest to replace "fact" with "conclusion/reasoning/hypothesis".
18. L440: At least for the two sites which feature classic radiation fog, Double and Half seemed to consistently increase and decrease certain aspects.
19. L464: "at THE microscale"
20. L465: "The occurrence of overlying clouds"
21. L476: "thorough"

---

## Editor Decision (ED1)

**Dear Dongqi Lin and co-authors**

Many thanks for your patience. Evaluating your paper is one of the most difficult cases I ever had as an editor, because of the strongly diverging opinions. Here I briefly summarize the status of the evaluation of your paper:

**First round**

Referee 1 was critical and made many constructive comments about how to improve the paper. Referee 2 was very critical and recommended to reject the paper. One important comment was "The numerical choices on the model configuration are not always clearly presented, and are sometimes debatable". This problem seems to remain, see below.

**Second round**

Referee 1 was positive about the improvements and recommended to accept the paper with minor revisions (which you implemented).

Referee 2 declined to do another review.

I invited a new Referee 3 who was very critical ("flawed methodology") and recommended rejection.

**Third round**

Referee 3 remains very critical. See their comments below. I therefore asked advice from two additional LES and fog experts. They agree with the limitations of your modeling approach, but overall, see value in your study for the fog modeling community. Below you find their comments (referees 4 and 5). Importantly, referee 4 makes very good suggestions to do some additional sensitivity tests.

**Decision**

There is clearly no consensus view from the experts about your paper, but it is time to soon conclude this long review process. And since three experts see value in your study, my decision is to accept your paper for publication in ACP, on the basis that you add a few sensitivity tests as suggested by reviewer 4. Thanks to the transparent review process of ACP, this difficult decision, and the reasoning behind, will be apparent to the community. I see my decision in the spirit of "in dubio pro reo". Some doubts remain about the model setup, but often in science, it is also valuable to publish results with limitations and caveats, as long as they are explicitly discussed in the paper.

Therefore, my request is that you submit a revised version of your paper where you

- 1) Check again that you explain as transparently as possible your decisions for the model setup and its limitations. The comments from reviewers 3-5 are certainly helpful for this.
- 2) Include additional sensitivity experiments (e.g., in the form of an Appendix), as suggested by reviewer 4.

With best regards, Heini Wernli

**Referees' comments in round 3**

**Referee 3**

I recommend rejection of the manuscript because the authors (and they confirm this) are misusing an LES model in a way which is simply not allowed. This is a severe technical flaw, but I do not see that the authors realize this.

The authors did address my major concern that they used a non-adequate grid spacing in PALM to simulate a fog event. However, they agree that the resolution is not sufficient and that they in fact did not perform an LES simulation. They provide some reasoning why they did not use a RANS model, but that does not make the methodology sound and safe. In fact, using an LES model (i.e. if most of the flow is parameterized by a subgrid-scale model which assumes that the flow is isotropic, turbulent and the eddies to be parameterized are in the inertial subrange of turbulence will simply not work. The results will be wrong. Point. The reasons that PALM does a better job in representing the complex surface does not count, since the flow dynamics are faulty. I thus have to recommend rejection.

**Referee 4**

Firstly, I do not agree with the expert (referee 3) that the results will simply be wrong because PALM is an 'LES' model being used outside of the LES regime. I'm not an expert in PALM, but if asked to construct a numerical model at the hectometric (~100m) scale it would require: A dynamical equation set capable of representing all important processes at that scale – I've no reason to suspect PALM does not have this.

A suite of physical parametrizations to represent unresolved processes – again, I've no reason to suspect PALM does not have this. In particular, the key issue is likely to be around the subgrid turbulence parametrization. The one they do use is perhaps not the best, but it's also not wrong. We'd want the scheme to be primarily local in nature (i.e. non-local fluxes required in NWP/climate models aren't needed at this scale), which it is, and the scheme definitely needs to be 3D (the 1D assumption in NWP/climate models is also not valid any more), which again it is. So it's fair to say that I do see merit in their study, and don't believe their choice of tool precludes publishing the study. This is the scale at which we're going to be doing NWP of fog in the coming years, so it is a valuable resolution for research.

However, I would also raise criticisms of the model setup that I don't believe have been ac

However, I would also raise criticisms of the model setup that I don't believe have been adequately addressed by the authors. In particular:

It would be important to characterise the sensitivity of their results to the subgrid mixing scheme they are using. If the scheme is just taken unaltered from LES scales, it's likely to generate a mixing length which is proportional to the model grid-length, rather than any physical scale. This is likely to be too large given their coarse resolution, therefore **it would be nice to see some alternative simulations** with an appropriately tuned scheme to give a more physical mixing length for stable boundary layer / fog conditions. I think this would help alleviate the other reviewers concern, particularly if it didn't change their overall conclusions.

I'm slightly concerned about their vertical grid setup – the key feature of NWP-style models is usually having a stretched vertical grid, to give enhanced resolution near the surface and properly represent the near surface processes. This can have a huge impact on fog simulation. 18m in the vertical feels way to coarse to me – I'd expect to see a lowest level around 1-2m and some sort of quadratic stretching away from this (maybe ending up at a uniform 18m when that is reached). So again, I'd probably **like to see some sensitivity test** of how vertical resolution affects their results.

Hope that helps! I'd be happy to act as a reviewer, but don't think I'd have much to add on top of what I've said above if I did read the full paper.

**Referee 5**

I agree with the reviewer and the authors that the simulations should not be termed "LES", but rather "high-resolution mesoscale simulations" (as indicated by the authors in the revised manuscript; following also the suggestion by Cuxart 2015).

This review seems to touch upon a long-standing debate between LES purists and more applied users of LES and mesoscale models, as for example, in the mountain meteorology community.

While the former rightly claim that the majority of the turbulence should be explicitly resolved in a LES (say 90% or more) in order to fulfill the basic assumption of LES. This is because the formulation of the LES subgrid-scale turbulence model assumes that the large eddies are well resolved and that only the universal inertial subrange of turbulence needs to be parameterized.

On the other hand, if you are interested in multiscale problems, which include, for example, local circulations due to topography or land surface heterogeneity in addition to the turbulent scales, then high-resolution mesoscale simulations are also of value. And due to computational constraints often the only realistic possibility, as larger domain sizes are required. The focus of these simulations is then typically to study the impact of the local circulations on the problem, taking into account a less accurate representation of turbulence.

As the current simulations are carried out in the turbulence grey zone, neither the assumptions of LES, nor those of RANS, are fulfilled. My first impulse, was also, why did the authors not use WRF in RANS mode. However, considering the authors arguments and that we are anyway in the turbulence grey zone, for which no universally accepted parameterizations exist, I can accept the authors reasoning for the current setup. In that case the LES closure can, in my opinion, be considered as a poor parameterization for the turbulence grey zone. A setup which is not unusual in the literature.

As long as the setup is clearly communicated and aptly named (i.e., high-resolution mesoscale simulations), I deem it acceptable.

---

## Author Response (AR2)

**Author's response for egusphere-2022-1229:**

**Investigating multiscale meteorological controls and impacts of soil moisture heterogeneity on radiation fog in complex terrain using semi-idealised simulations**

The authors thank the reviewers and the editor for their time and careful consideration given to this manuscript. We agree with the major concerns from Reviewer #3 regarding the configuration of Large Eddy Simulations (LES), while we would like to further clarify and emphasise here that we only performed high-resolution mesoscale simulations for the experiments presented in the manuscript. On reflection, it is clear that our simulations are actually not, and should not be called, LES. We have therefore now followed the terminology described by *Cuxart (2015) – 'When Can a High-Resolution Simulation Over Complex Terrain be Called LES?'* We have acknowledged in the manuscript that, with the grid spacing we have used in our simulations, we did not resolve large eddies and hence our simulations should not be called LES. In addition, there are several practical reasons why we are not able to use Reynolds-averaged (Navier–Stokes) (RANS) simulation models as suggested by the editor, and we have outlined these in detail below in response to Reviewer #3's major comment 1.

It should be noted that the main focus of this manuscript is not to replicate a fog event accurately, resolve turbulence, or investigate the impact of turbulence in fog. To properly resolve large eddies in a stable boundary layer, the grid spacing needs to be finer than at least 4 m, which is extremely computationally expensive considering the domain size of the simulations. The purpose of our configuration is to carry out experiments on soil moisture heterogeneity quickly, with the heterogeneity of terrain and other surface structures included in the simulation. It is clear to us that with a grid spacing finer than 4 m, such experiments on soil moisture heterogeneity are not computationally feasible.

We understand the limitations of our approach, although it is clear that we did not explain them in sufficient detail in the previous versions of the manuscript. This appears to have diverted discussion of the manuscript to focus on whether proper LES configuration was used. We have therefore added additional description to clarify that our simulations are high-resolution mesoscale simulations rather than LES. We have also added more discussion regarding the limitations of our approach.

The reviewer's comments have been listed below in *black italics* and responded to individually in *blue italics*. Revised sentences are in *red italics*.

**Reply to Stephanie Westerhuis:**

- L18 and 38: As the model's horizontal resolution is on the order of several decametres, I would rather specify the microscale as "on the order of 100m to 1km". We have changed "(1 cm to 1 km)" in L18 to "on the order of 100 m to 1 km" as this is what this study has shown. We did not change this for L38 as it is for the broader concept of microscale in the introduction.
- 2. L159: "This means THAT..." This has been corrected.

3. L178: It is not self-explanatory what is meant by a "3D profile in west-east, south-north, and the vertical direction". I would replace the term "profile", as this in itself is associated with an extract of a quantity along the vertical dimension. Do you not just mean "soil moisture which varies both in the horizontal and vertical"? Yes, this means "soil moisture which varies both in the horizontal and vertical dimensions". We have replaced the term "profile" with "field" as follows:

... a 3D field of soil moisture (west-east, south-north, and eight vertical soil layers) ...

- 4. L193 and 198: Repetition of "bulk cloud model only enabled in D04." We have removed "The bulk cloud model was only applied in D04." in L198.
- 5. L199: "Cloud water sedimentation IS based on..." We have revised this as follows:

Cloud water sedimentation based on Ackerman et al. (2009) is enabled.

6. L213: Improve wording of "fog event is the most significant". We have revised this as follows:

We focus on four sites in D04 where fog events are the most recognisable and have a relatively long duration.

- L223: Swap ending of sentences: "..cross sections. The sunset time...on Day 1 and sunrise..." This has been corrected.
- 8. L227: Either "temperature decreases" or "air cools" We have revised this as follows:

... where the temperature decreased faster ...

- 9. L237: no "9" in "03900 LST". This has been corrected.
- 10. L253: "High qv was PRESENT..." This has been corrected.
- 11. L254 263: I find the explanation that clouds instead of fog formed very convincing and would replace "This is suspected to be due to" with "This is likely caused by". I would however specify the clouds to be "low stratus" which is often related to fog occurrence. The physical drivers could be more precise: "The layer of low stratus reflecting the outgoing LW radiation results in a reduced surface cooling and.." E.g. on the Swiss Plateau, the transition from fog to low stratus (and back) is often associated to increase (decrease) in wind speed. Is this also the case here? We have revised this as follows:

This is likely caused by the layer of low stratus that formed over this area at around 400 m above the ground, while the southern part of the simulation domain was under clear sky conditions (see Figure 6i). The formation of the low stratus layer is considered to result from high qv at around 400 m at model initialisation (Figure 2c), with high values of qv at approximately 400 m also visible in Figures 6e-h. The layer of low stratus absorbed the outgoing longwave radiation from the surface and re-radiated it to the surface over the northern section of D04, which is likely to have led to reduced surface cooling and subsequently a less stable near-surface layer over this area.

Regarding the transition between fog and low stratus, we do think that higher wind speed is responsible for the formation of low stratus instead of fog in the north part of the domain. The processes involved in this transition are interesting. However, as this is out of the scope of this study, we do not discuss it further. Future work may be carried out to investigate this topic in greater depth.

12. L263: Suggestion: "Besides the low stratus clouds the simulation featured clear-sky conditions."

This has been corrected.

- 13. L331: T missing in "HET12p" This has been corrected.
- 14. Figure 11: Specify not only the black solid line but also the grey shading in the first two columns.

We have added a description for the grey dots in Figure 11 as follows:

In the first two columns, the grey dots are obtained from all grid points at 1 km and 3 km scales, respectively.

- 15. L389: "Develop INto" This has been corrected.
- 16. L408 and 409: The reasoning is not logically consistent: a) High LHF = high QL b) High soil moisture = high LHF c) High soil moisture =/ high QL -> Probably a) should be phrased differently.

As shown in Figure 12a, we can see that a) high LHF = high QL and b) high soil moisture = high LHF for fixed values of QL, but c) high soil moisture  $\neq$  high QL, which leads to the discussion in L445 that advection of water vapour in the atmosphere plays a more important role compared to evaporation from the soil. We understand that the logic here could be confusing and have rephrased this as follows:

As shown in Figure 12a, at HAP, higher  $LHF_{Acc}$  is generally associated with higher  $qI_{Acc}$ . In addition, although higher soil moisture shows a positive correlation with  $LHF_{Acc}$  for fixed values of  $qI_{Acc}$ , an increase in soil moisture does not coincide with a higher  $qI_{Acc}$ .

- 17. L420: I suggest to replace "fact" with "conclusion/reasoning/hypothesis". We have replaced "fact" with "conclusion".
- 18. L440: At least for the two sites which feature classic radiation fog, Double and Half seemed to consistently increase and decrease certain aspects. We agree with the reviewer and have added the following discussion:

Nevertheless, the changes in fog occurrence, formation time, and dissipation time do not show a linear correlation with changes in soil moisture across the domain. At HAP and PTH, where fog reflected more localised processes, fog duration decreased more than 20 minutes when soil moisture was halved. However, this relationship is less clear for SWC and WMR.

- 19. L464: "at THE microscale" This has been corrected.
- 20. L465: "The occurrence of overlying clouds" This has been corrected.

**21. L476: "thorough"**

We have revised this as follows:

... requires further research through, for example thorough additional case studies, as only one case study is presented here.

**Reply to Anonymous Referee #3:**

**Major comments:**

1. A four-step multiple self-nesting of PALM is employed, with grid spacings between 729 m and 81 m horizontally and 162 m to 18 m vertically. As outlined in the general summary: none of these grid spacings are sufficient to resolve the turbulence in a typical environment prone to radiation fog. Even for a convective boundary layer, where the dominant eddies are large, the coarsest grid spacing allowed is around 100 m. The grid spacings used here in the domains D01-D04 are way beyond what is possible to use in an LES model. Under stable conditions, the largest eddies are usually not larger than 10 m, so the grid spacing must be way smaller than that. You either need much higher grid spacings (in LES of radiation fog, grid spacings in literature are in the order of 1 - 4 m horizontally and vertically!), or you need to use a RANS model. By violating the constraints of LES, you are parameterizing all turbulent transport with a subgrid-scale model, which assumes to only treat small-scale isotropic turbulent fluxes. As this is not the case, the transport will be totally wrong. There is no discussion about this in the paper, except one sentence, saying that most of the turbulence is parameterized. Furthermore, no vertical profiles and turbulent quantities are presented. One might suspect this is because they will immediately show these flaws. If the authors cannot correct for these flaws, they probably better go for a RANS model where the grid spacing issue is somewhat less severe (however, to resolve fog layers, small vertical grid spacings are still essential!).

As we mentioned in the general summary above, our simulations are not LES, and should be called "high-resolution mesoscale simulations". After inspecting the model results, the maximum ratio of resolved TKE in our simulations is only around 35%, which confirms the reviewer's concerns and affirms the need to call the experiments high resolution mesoscale simulations. We did not show vertical profiles and turbulent quantities because the main focus of this study is the impact of spatial heterogeneity in soil moisture on fog in a complex environment. If one aims to simulate radiation fog and understand the impacts at the microscale with LES, then we do agree that the grid spacing should be finer.

There are several reasons why we were unable to use a RANS model:

1) The RANS code in PALM is not fully implemented for the application of fog and several users including the PALM developers have reported that RANS in PALM does not improve the computation time (based on personal communication and PALM ticket system, e.g. https://palm.muk.uni-hannover.de/trac/ticket/1444; as login credentials are required to access PALM ticket system, a screenshot is attached below). In this study, we aim to have multiple experiments without excessively using computational resources. Therefore, using RANS in PALM is not useable in our study.

Please note that the way the TKE-e and TKE-l closures are implemented, the simulation will most likely not be faster compared to the LES approach. Certain optimizations are not implemented in the RANS mode so far that would allow for a larger time step and, hence, a faster integration with regards to computing time.

There is no fixed plan when these turbulence closures will be further developed and optimized, yet. So, it might take some time until this mode offers a considerably faster alternative to the LES mode.

2) An alternative suggestion could be to use other RANS models such as WRF. However, WRF does not include and resolve surface heterogeneity well compared to PALM. PALM offers more features in its land surface modules, which has enabled us to conduct the simulations we have presented. In addition, previous studies such as Cui et al. (2019) compared WRF with WRF-LES at the same grid spacing in radiation fog and showed that WRF-LES has better performance with the advent of resolving fluctuations in the state parameters that subsequently reduced the mean bias when compared to observations. Please note that in Cui et al. (2019) their finest grid spacing is 333.33 m, which should also not be called LES. Despite this limitation, their simulation results still show good agreement with observations, demonstrating that using LES model at a coarse grid spacing (i.e., as a high-resolution mesoscale model) still has practical value.

This is why we decided to conduct high-resolution mesoscale simulations using the PALM LES code. We discuss the reason why we have chosen this approach in the manuscript (L56-90) as follows:

Considering the high computational cost, the optimal approach is to carry out highresolution mesoscale simulations (Cuxart, 2015) at sub-km grid spacing. The surface and topographic heterogeneities can be partially resolved in such high-resolution mesoscale simulations, and consequently the dynamical processes and spatial variability of fog can be captured (Vosper et al., 2013, 2014). This study therefore aims to investigate the impact of soil moisture on radiation fog duration using high-resolution mesoscale simulations for Christchurch.

We agree with the reviewer that we did not provide sufficiently detailed discussion on the limitations of our approach, which could have generated confusion among the LES community. We have further clarified the terminology relating to our approach in Section 3 as follows:

Therefore, following the terminology discussed by Cuxart (2015), our simulations are high-resolution mesoscale simulations rather than LES, despite using the LES model PALM.

In addition, we have added discussion regarding this limitation as follows:

Due to the difficulty in spatial analysis and the significant computational cost, we only carried out simulations at a horizontal grid spacing of 81 m. In our high-resolution mesoscale simulations, most of the eddies are not resolved. With PALM's high scalability at microscale, the grid spacing of the simulations should typically be finer so that turbulence structures can be better resolved and captured.

Furthermore, the reviewer mentioned in their general summary that:

In the discussion section they report different results than found by a previous study (Maronga & Bosveld, 2017), both using the same PALM model system. It is likely that these differences are simply due to insufficient grid resolution. As was pointed out by Maronga & Bosveld, the required grid spacing for a typical radiation fog event was 1 m (both vertically and horizontally).

The difference between this study and Maronga and Bosveld (2017) has been discussed in Section 5.3. We believe the main reason for the difference is that our simulations include different types of fog, while Maronga and Bosveld (2017) only simulated radiation fog. As reviewer Dr Stefanie Westerhuis mentioned in her comment #18, for the two sites that experienced classic radiation fog, doubled and halved soil moisture do show consistent changes in some fog characteristics. For example, as shown in Figure 10a, at HAP, Half did not lead to significant changes in formation time, but dissipation time was affected. This agrees with Maronga and Bosveld (2017). We understand that the study of Maronga and Bosveld (2017) showed that to replicate the vertical structure and duration of a historic fog event, the grid spacing should be as fine as 1 m. It should be noted that they also have pointed out that this could be extremely computationally expensive. In this manuscript, we do not aim to forecast fog or to replicate any historic fog events. As we did not discuss the difference in grid spacing between the two studies, we have added the discussion as follows:

It should be noted that this study uses a coarse grid spacing (horizontal grid spacing of 81 m) compared to Maronga and Bosveld (2017) (horizontal grid spacing finer than 4 m). Our simulations did not resolve large eddies and hence the turbulence transport could be expected to differ significantly. However, running simulations over an area of approximately 17.5 km × 17.5 km with grid spacing finer than 4 m is not computationally feasible. Future work should therefore be carried out using a finer grid spacing when suitable computation resources become available.

2. You report you are using RRTMG as radiation code, but you also refer to have complex terrain and buildings in the domain. As RRTMG is operating as a single vertical column model, how do you calculate radiative fluxes at non-horizontal surfaces? As far as I know, PALM automatically uses a radiative transfer scheme (RTM) as soon as buildings or complex terrain is found in the domain. RTM, however, cannot consider clouds and only works for clear-sky conditions. Also, it does not calculate flux divergences, which play a key role in fog development. I found no statement on how this problem is treated in the study.

As we have a plant canopy and an urban canopy with the microphysics module switched on, the RTM in PALM is required to be switched off. Details can be found in PALM documentation in this link:

https://palm.muk.unihannover.de/trac/wiki/doc/app/radiation\_parameters#radiation\_interactions\_on which presents the following warning:

**Warning:** radiation\_interactions\_on = .T. is not allowed, in case the bulk cloud model (BCM) is used together with the urban- and/or land-surface model (USM and/or LSM) and the radiation model.

Due to the aforementioned model limitation we have to neglect the radiative transfer processes within the canopy, including shade and reflections which RTM usually perform. Therefore, no 3D RTM was used in our simulations. We understand that this means that 3D features, such as shadows, are not included in the simulations. However, as the reviewer stated, RTM neglects the absorption, scattering and thermal emission within the air mass and hence is not suitable to use for fog simulations. In addition, in PALM, RTM is only applied from the lowest model level to the top of the highest surface obstacle (such as buildings or plant canopy), and above that height, the atmosphere uses a radiation model like RRTMG to calculate radiation. If RTM is switched on with RRTMG, then the exchange between surface radiation and the atmosphere is controlled by RTM. The surface radiation passed from RTM to RRTMG uses averages and therefore RRTMG does not see any surface heterogeneity, which could be a potential downside for studies that focus on the effects of heterogeneous features on the surface. For more details, please refer to Krč et al. (2021) and Salim et al. (2022).

In our simulations, we have utilised the RRTMG with the RTM deactivated. This allows the calculation of radiation to include the heterogeneous surface properties. Although this approach results in the absence of shadows in the simulations during daytime, this limitation is consistently applied across all our simulations. Furthermore, our study exclusively focuses on the nocturnal development of fog, during which the presence of shadows is not relevant.

We also would like to highlight that the RRTMG employed in our study is consistent with the methodology utilised in the PALM fog studies conducted by Maronga and Bosveld (2017) and Schwenkel and Maronga (2019). As these studies specifically focused on fog simulations without incorporating plant canopy or urban surfaces, the use of an RTM was not necessary. Nevertheless, their findings demonstrated that RRTMG is suitable for simulating fog. While our primary objective is not to precisely replicate all processes involved in the fog lifecycle, we acknowledge that the choice of radiation schemes in our research may present limitations. In response to this concern, we have added a detailed description of the radiation model in Section 3 of the manuscript as follows:

Due to the inclusion of the plant canopy model and bulk cloud model in PALM, the threedimensional Radiative Transfer Model (RTM; Krč et al., 2021) was switched off. The surface radiation transfer is then directly computed by the RRTMG model embedded in PALM. This configuration of the radiation model in PALM fog simulations is similar to those described in Maronga and Bosveld (2017) and Schwenkel and Maronga (2019).

And we have added discussion on this limitation in Section 6 as follows:

Furthermore, our simulations only used the RRTMG scheme in PALM for computation of radiation and did not include the three-dimensional RTM. As discussed in Salim et al. (2022), choices of the radiative transfer processes included in PALM can change the flow field considerably in an urban environment. The RTM applied in PALM neglects the absorption, scattering, and thermal emission by air masses (Krč et al., 2021; Salim et al., 2022), and hence its application is limited in case of fog simulations. A recent ongoing development regarding the three-dimensional radiative transfer in PALM is the implementation and integration of the TenStream radiative transfer model. TenStream is capable to consider the effects of three-dimensional radiative transfer on the atmospheric heating rates or dynamic heterogeneities such as moving clouds or fog (Jakub and Mayer, 2015, 2016). With such development and implementation of the radiation model in PALM, TenStream should be considered and utilised for future fog simulations.

3. Why is the most simply cloud microphysics available in PALM used? By default, PALM uses a two-moment scheme, which is kind of a standard for years. Is there any reasonable argument for switching to a simplistic Kessler scheme? Furthermore, cloud physics are only allowed in the D04 domain, which means that fog cannot be advected in the D04 domain. Does that make sense? Also, this means that there can be supersaturated air inside the D01-D03 domains. If this air is advected into D04 it will lead to spurious condensation.

As we mentioned in Line 198 of the manuscript, "Two-moment schemes are not compatible with the plant canopy model of PALM" (see also PALM error message here https://palm.muk.uni-hannover.de/trac/wiki/doc/app/errmsg#PA0360). In our simulations, we included plant canopy and hence were not able to use two-moment schemes. Schwenkel and Maronga (2019) indicated that the general fog life cycle simulated using a one-moment scheme is acceptable. However, ideally, we do agree that one would aim to use a two-moment scheme, which performs better in simulating fog as stated in e.g., Schwenkel and Maronga (2019). We have added discussion on this limitation as follows:

Due to the inclusion of the plant canopy, only a one-moment microphysical scheme was used in this study. Although, as described in Schwenkel and Maronga (2019), the use of a one-moment scheme does not affect the general structure of fog life cycle, future work may aim to apply two-moment microphysical schemes for a more realistic representation of the microphysics.

We only enabled microphysics in the D04 domain because we wanted to simplify the processes involved in the fog events that we simulated. We do not aim to forecast or replicate fog events. Rather, we aim to investigate the impact of soil moisture heterogeneity on fog at the surface. The processes involved in the current simulations are already complex to analyse, as discussed in Section 5. If the microphysics were switched

on for the parent domains, analysis of the processes related to soil moisture heterogeneity would be very difficult. We have added the following clarification in Section 3:

The bulk cloud model was switched off for domains D01, D02, and D03, to simplify the processes involved in the simulated fog.

4. The authors did a purely idealized study with no relation to any observed fog case. While I would agree that this might not be overly critical, in this particular case it makes me worry. As a combination of the technical flaw, the reader cannot evaluate whether the obtained results are by any means realistic.

We would like to point out that we carried out our simulations using data obtained from an observed fog case and WRF simulations as described in Section 2.2 (Lines 135-141), Section 3 (Line 169-172), and Appendix A. Furthermore, as we stated in our previous response to Reviewer #2, the initialisation profiles were carefully selected so that radiation fog can be simulated in PALM. We carried out a comparison between WRF and the observations regarding the vertical profiles of winds and temperatures as shown in the figures below:

In these figures, 'sodar' indicates observations obtained from the Sound Detection and Ranging (SoDAR) wind profiler deployed at the airport, 'AWS' indicates data obtained from the automatic weather station (AWS) located near the sodar (temperature at 1.25 m and wind speed and direction at 10 m), 'T2' is the air temperature at 2 m simulated by WRF, and 'WRF Tair' indicates WRF air temperature at various heights. The upper air observations were obtained from the national climate database (CliFlo; https://cliflo.niwa.co.nz/) operated by the National Institute of Water and Atmospheric Research (NIWA) include two sets of data: one named as temperature data set, and one named as wind data set. These upper air measurements were recorded by sensors on aircraft arriving at and departing from Christchurch airport. The CliFlo temperature data set includes vertical profiles of temperature only, while the CliFlo wind data set includes vertical profiles of wind in addition to temperature. In general, WRF shows good agreement with all the observations.

We agree with the reviewer that the simulations presented in this study are to some extent idealised. We did not present any validation of the simulations against observations as we did not aim to replicate the fog event accurately. However, we believe that this does not rule out the value of our study. When comparing the simulated results to observations, the PALM simulations show better performance than the WRF simulations. Figures AC1-AC4 below show the comparison between observations and WRF and PALM simulations. The observational data were obtained from the AWS and the sodar located at Christchurch International Airport. As shown in Figure AC1, WRF consistently overestimates temperature. PALM follows the trend presented in WRF, but exhibits a smaller temperature bias. The time series of 10 m wind speed shown in Figure AC2 also show agreement between WRF, PALM, and the observations. At 50 m above ground level (AGL), WRF highly overestimated the wind speed towards the end of the simulation period while PALM still shows quite good agreement with sodar observations (Figure AC3). In addition, PALM-simulated wind speed anomalies agree with the sodar observed wind speed anomalies at 30 m AGL as shown in Figure AC4, despite an underestimation of wind anomalies in PALM. The wind anomalies are calculated by subtracting the instantaneous wind speed from the hourly averaged wind speed for each

hour during the period between 2000 LST 5th August 2001 and 0000 7th August 2001. The temporal frequency of PALM output is 1 minute while the sodar data were obtained every 10 minutes, which could be one of the explanations regarding the underestimation in PALM. The wind anomaly statistics of WRF are not shown here because the WRF simulation we have used here only has hourly output and only the average properties of airflows are presented in WRF simulations with the RANS mode.

---

## Author Response (AR3)

**Authors' response for egusphere-2022-1229:**

**Investigating multiscale meteorological controls and impacts of soil moisture heterogeneity on radiation fog in complex terrain using semi-idealised simulations**

We want to thank the editor and all the five reviewers for their time and consideration given to this manuscript. We understand the reviewers' and the editor's concerns regarding the grid spacing configuration of the simulations presented in the manuscript.

We acknowledge that the grid spacing used in this study falls in the "Grey Zone" of boundary layer modelling, which, as Reviewer #5 mentioned, has been a long-standing debate in the community. We understand the limitations of our current configuration, but decided to use PALM because it has the potential and high scalability to run the fog simulations at finer grid spacings. During the review process, we revised our model setup description. As mentioned by Reviewers #4 and #5, running simulations with fine grid spacing (finer than 10 m) is computationally expensive. Our configuration aims to conduct experiments quickly (although it could be less accurate) and to potentially provide additional insight into numerical weather prediction (NWP) of fog.

As discussed by the editor and Reviewer #4, to provide more confidence for our simulations and results, we have conducted sensitivity experiments on the vertical grid spacing and have presented the results in the supplements in this revision. In addition, we have revised the description of the model setup and the limitations of our configuration.

The editor's comments have been listed below in *black italics* and responded to individually in *blue italics*. Revised sentences are in *red italics*.

*1) Check again that you explain as transparently as possible your decisions for the model setup and its limitations. The comments from reviewers 3-5 are certainly helpful for this.*

*Along with the sensitivity test, we have provided an additional explanation for using the configurations in our simulations in the discussion:*

*Our simulations did not resolve large eddies, and hence, the turbulence transport could be expected to differ significantly from Maronga and Bosveld (2017). The grid spacing used in this study falls in the "Grey Zone" of turbulence (e.g., Honnert et al., 2020), which allows us to carry out multiple experiments relatively quickly over a large area, although the representation of the turbulence is less accurate. To provide more confidence in our simulations, we have carried out sensitivity experiments on the vertical grid spacing of D04 (from 18 m to 9 m and 6 m). Our conclusions remain unchanged when a finer vertical grid spacing is used (see supplements). In addition, running simulations over an area of approximately 17.5 km × 17.5 km with grid spacing finer than 4 m is not computationally feasible. Future work should therefore be carried out using a finer grid spacing when suitable computation resources become available.*

*2) Include additional sensitivity experiments (e.g., in the form of an Appendix), as suggested by reviewer 4.*
*We have conducted several sensitivity experiments and decided to attach the sensitivity experiments as supplements. The entire sensitivity experiments are also attached at the end*

*of this response. The sensitivity experiments are shown in Figures S1-S6. The results of the homogeneous simulation with simulation domain D04 of 18 m vertical grid spacing conducted using the Intel FORTRAN compilers are presented in Figures S7-S10.*

*We conducted the fog simulations presented in the manuscript using the Cray Fortran compiler, which is no longer compatible with PALM revision 4829 after the major upgrade of our supercomputing facility earlier this year. For the sensitivity experiments presented here, we have used the Intel Fortran compiler instead. We acknowledge that the simulated results differ from those presented in the manuscript. Regarding the occurrence of fog, approximately 84.2% of the grid points remain unchanged. Around 7.5% of the grid points have fog simulated with the Cray Fortran compiler but do not have fog simulated when the Intel Fortran compiler is used. The remaining 8.3% of the grid points have the opposite results (have fog with the Intel compiler while no fog with the Cray compiler). Fog duration has reduced over the Port Hills area (PTH) by over 50 minutes, while the four locations Hagley Park (HAP), PTH, Southwest of Christchurch (SWC), and Waimakariri River (WMR) are still associated with the most recognisable fog events. This, however, does not change the conclusions of this study. To prove this, we conducted two fog simulations with the original domain configuration for HOM (homogeneous soil moisture) and HET12p (12-point adjustments of soil moisture heterogeneity).*

*In the sensitivity experiments, we have configured Domain 4 with a vertical grid spacing of 9 m and 6 m for HOM and HET12p. Reviewer 4 mentioned stretching of vertical grid spacing with the lowest level of around 1-2 m. However, this cannot be done in PALM with domain nesting for the time being. The computational cost increases dramatically with a finer vertical grid spacing. Hence, we cannot conduct a fog simulation with vertical grid spacing finer than 6 m within a sensible time frame. We have tried to run simulations with Domain 4 of 3 m vertical grid spacing. However, in our Cray XC50 supercomputing cluster, a 1-hour simulation of 3 m vertical grid spacing took over 6 hours of wall clock time on 369 Intel Xeon Skylake Gold 6148 processor cores, running at 2.4 GHz, meaning the entire simulation (48 hours / 2 days) will take at least 288 hours (12 days), which is not allowed with the enforced job time limit of 24-hour wall clock time. In our Cray CS400 supercomputing cluster, a 34-hour 6 m vertical grid spacing simulation took approximately 72 hours of wall clock time on 369 Intel Broadwell processer cores running at 2.1 GHz. The enforced job time limit for the Cray CS400 cluster is three days of wall clock time, meaning we cannot use this configuration for simulations with 3 m vertical grid spacing. We have tested other partitions in our cluster, for example, on the AMD EPYC Milan 7713 processer cores (2.0 GHz base clock, 3.675 GHz max boost clock), but*

1) *PALM has MPI compatibility issues on these CPUs, and we cannot scale up the number of CPUs used;*
2) *A 2-minute simulation with 3 m vertical grid spacing took 6 hours of wall clock time on 244 AMD processer cores.*

*Therefore, we decided to only conduct the sensitivity tests with a vertical grid spacing of 9 m, and 6 m. Our conclusions remain the same based on the sensitivity experiment results. In general, the change of vertical grid spacing does change the areas where fog occurred but does not alter our conclusions that the variation in soil moisture heterogeneity does not show a clear correlation with the change in fog duration.*

**Supporting information for "Investigating multiscale meteorological controls and impacts of soil moisture heterogeneity on radiation fog in complex terrain using semi-idealised simulations"**

This document presents sensitivity experiments on the vertical spacing used for Domain 4 (D04) of the fog simulations presented in the main manuscript. The original simulation has a vertical grid spacing of 18 m for D04, and we conducted two sets of simulations with a vertical grid spacing of 9 m and 6 m, respectively. The sensitivity experiments are presented in Section 1 and Figures S1-S6. The original simulations presented in the manuscript were conducted using the Cray FORTRAN compilers, while the sensitivity experiments presented here were conducted using the Intel FORTRAN compilers. This does not change our conclusions. To provide more confidence, the results of the homogeneous simulation with D04 of 18 m vertical grid spacing conducted using the Intel FORTRAN compilers are presented in Section 2. The figures (Figures S7-S10) in Section 2 are similar to those in Section 4 of the main manuscript.

**1. SENSITIVITY EXPERIMENTS**

The vertical level configurations and computation time of the sensitivity experiments are shown in Table S1. For each of the three configurations, two simulations were conducted. One contains homogeneous soil moisture (HOM), and the other contains heterogeneous soil moisture with the 12-point adjustment (HET12p). Hereafter, we denote the simulations conducted with the vertical grid spacing of 18 m, 9 m, and 6 m as HOM-18m and HET12p-18m, HOM-9m and HET12p-9m, and HOM-6m and HET12p-6m, respectively.

The fog duration for HOM-18m, HOM-9m, and HOM-6m, and the fog duration difference between each HOM simulation and HET12p simulation are shown in Figure S1. As shown in Figure S1, the finer vertical grid spacing results in an increase in fog area and fog duration. However, the comparison between each HOM and HET12p simulation still does not show a direct correlation between spatial variations in soil moisture and the changes in fog duration. To provide more evidence supporting our conclusion, Figure S2 shows the time series of liquid water mixing ratio ($ql$) for Hagley Park (HAP), Port Hills (PTH), Southwest of Christchurch (SWC), and

**Table S1.** Vertical level configurations and computation time for the sensitivity experiments.

| Vertical grid spacing (dz) | Vertical grid points (nz) | Simulation hours | CPU specification | Number of CPUs | Wall clock time |
|---|---|---|---|---|---|
| 18 m | 36 | 48 hours | Intel Xeon Skylake Gold 6148 processor cores (2.4 GHz) | 396 | 8 hours |
| 9 m | 72 | 36 hours | Intel Xeon Skylake Gold 6148 processor cores (2.4 GHz) | 396 | 24 hours |
| 6 m | 108 | 34 hours | Intel Broadwell processer cores (2.1 GHz) | 396 | 72 hours |

Waimakariri River (WMR). The vertical profiles of wind, potential temperature, and $ql$ are shown in Figures S3, S4, S5, and S6 for HAP, PTH, SWC, and WMR, respectively.

[Figure]

**Fig. S1.** (a), (b), (c), and (d) show fog duration in minutes for HOM-18m, HOM-9m, and HOM-6m, respectively. (b), (d), and (f) are fog duration differences in minutes between each HOM and HET12p simulation for the simulations with a vertical grid spacing of 18 m, 9 m, and 6 m, respectively. Hagley Park (HAP), Port Hills (PTH), Southwest of Christchurch (SWC), and Waimakariri River (WMR) are marked by dots in each panel.

**2. RESULTS FOR THE HOMOGENEOUS SOIL MOISTURE SIMULATION WITH THE INTEL FORTRAN COMPILER**

Figures S7, S8, S9, and S10 show the results for HOM-18m simulated using the Intel FORTRAN compiler.

[Figure]

**Fig. S2.** Time series of liquid water mixing ratio ($ql$) for (a) HAP, (b) PTH, (c) SWC, and (d) WMR.

[Figure]

**Fig. S3.** Vertical profiles of the u component of winds (a–e), v component of winds (f–j), w component of winds (k–o), potential temperature (p-t), and *ql* (u-y) for HAP taken from HOM-18m, HOM-9m, and HOM-6m at the times indicated in the figures.

[Figure]

**Fig. S4.** Similar to S3, but for PTH.

[Figure]

**Fig. S5.** Similar to S3, but for SWC.

[Figure]

**Fig. S6.** Similar to S3, but for WMR.

[Figure]

**Fig. S7.** Similar to Figure 3 in the main manuscript, but for HOM-18m simulated using the Intel FORTRAN compiler.

[Figure]

**Fig. S8.** Similar to Figure 4 in the main manuscript, but for HOM-18m simulated using the Intel FORTRAN compiler.

[Figure]

**Fig. S9.** Similar to Figure 5 in the main manuscript, but for HOM-18m simulated using the Intel FORTRAN compiler.

[Figure]

**Fig. S10.** Similar to Figure 6 in the main manuscript, but for HOM-18m simulated using the Intel FORTRAN compiler.

---

## Author Response (AR4)

Dear Heini Wernli,

Thank you very much for the time, consideration, and substantial effort you have invested in helping us through this review process. We have included acknowledgements for the reviewers and you as the handling editor. The improvements made to the paper would not have been possible without their and your invaluable help. We have reviewed the reference list, addressing issues with the DOIs, and removing the publishers as requested.

We understand that there may be concerns and/or doubts regarding the reliability of the simulations. Fog in its nature is complex and sensitive. Our primary goal is not to achieve accurate fog simulations in this paper. Validation of the simulation results is crucial for future research in this area. We hope our paper will provide insights and inspiration for future work on radiation fog simulations.

Many thanks,

Dongqi Lin, Marwan Katurji, Laura E. Revell, Basit Khan, and Andrew Sturman